# Breaking the Expressive Bottlenecks of Graph Neural Networks

## Abstract

Recently, the Weisfeiler-Lehman (WL) graph isomorphism test was used to measure the expressiveness of graph neural networks (GNNs), showing that the neighborhood aggregation GNNs were at most as powerful as 1-WL test in distinguishing graph structures. There were also improvements proposed in analogy to $k$-WL test ($k > 1$). However, the aggregators in these GNNs are far from injective as required by the WL test, and suffer from weak distinguishing strength, making it become expressive bottlenecks. In this paper, we improve the expressiveness by exploring powerful aggregators. We reformulate aggregation with the corresponding aggregation coefficient matrix, and then systematically analyze the requirements of the aggregation coefficient matrix for building more powerful aggregators and even injective aggregators. It can also be viewed as the strategy for preserving the rank of hidden features, and implies that basic aggregators correspond to a special case of low-rank transformations. We also show the necessity of applying nonlinear units ahead of aggregation, which is different from most aggregation-based GNNs. Based on our theoretical analysis, we develop two GNN layers, ExpandingConv and CombConv. Experimental results show that our models significantly boost performance, especially for large and densely connected graphs.

## 1 Introduction

Graphs are ubiquitous in the real world. Social networks, traffic networks, knowledge graphs, and molecular structures are typical graph-structured data. Graph Neural Networks (GNNs) (Scarselli et al., 2008; Gori et al., 2005), leveraging the power of neural networks to graph-structured data, have a rapid development recently (Kipf & Welling, 2016; Hamilton et al., 2017; Bronstein et al., 2017; Gilmer et al., 2017; Duvenaud et al., 2015).

Expressive power of GNNs measures their abilities to represent different graph structures (Sato, 2020). It decides the performance of GNNs where the awareness of graph structures is required, especially on large graphs with complex topologies. The neighborhood aggregation scheme (or message passing) follows the same pattern with weisfiler-lehman (WL) graph isomorphism test (Weisfeiler & Leman, 1968) to encode graph structures, where node representations are computed iteratively by aggregating transformed representations of its neighbors with structural information learned implicitly. Therefore, the WL test is used to measure the expressiveness of GNNs. Unfortunately, general GNNs are at most as powerful as 1-order WL test (Morris et al., 2019; Xu et al., 2019). There is also work trying to improve the expressiveness that are beyond 1-order WL test (Maron et al., 2019; Morris et al., 2019; Chen et al., 2019; Li et al., 2020b; Vignac et al., 2020). However, the weak distinguishing strength of aggregators is the fundamental limitation. The expressiveness analysis measured by the WL test assumes that aggregators are injective, which is usually unreachable. Therefore, this motivates us to investigate the following questions: What are the key factors to limit the expressiveness of GNN? and how to break these limitations?

Aggregators are permutation invariant functions that operate on sets while preserving permutation invariance. (Zaheer et al., 2017) first theoretically studied permutation invariant functions and provided a family of functions to which any permutation invariant function must belong. (Xu et al., 2019) extended it on multisets but only for countable space. (Corso et al., 2020) further extended it to uncountable space. (Murphy et al., 2018) and (Murphy et al., 2019) expressed a permutation invariant function by approximating an average over permutation-sensitive functions with tractability

strategies. (Dehmamy et al., 2019) showed that a single propagation rule applied in general GNNs is rather restrictive in learning graph moments (Lin & Skiena, 1995). They and (Corso et al., 2020) improved the distinguishing strength of aggregation by leveraging multiple basic aggregators (SUM, MEAN, NORMALIZED MEAN, MAX/MIN, and STD). This strategy showed its effectiveness on tasks taken from classical graph theory.

In contrast to existing studies towards aggregators in GNNs, we provide a new GNN formulation, where the aggregation is represented as the multiplication of the corresponding hidden feature matrix of neighbors and the aggregation coefficient matrix. This new formulation enables us to answer the following questions: (i) when a GNN will lose its expressive power; (ii) How to build aggregators with higher distinguishing strength, even injective aggregators. Based on our theoretical analysis, we propose two GNN layers: ExpandingConv and CombConv, and evaluate them on general graph classification and graph regression tasks. Our key contributions are summarized as follows:

- We formalize the distinguishing strength of aggregators as a partial order, and theoretically show that the choice of aggregators can be bottlenecks of expressiveness. We also propose to apply nonlinear units ahead of aggregations to break the distinguishing strength limitations of aggregators as well as to achieve an implicit sampling mechanism.
- We reformulate the neighborhood aggregation with the aggregation coefficient matrix and then provide a theoretical point of view on building powerful aggregators and even injective aggregators.
- We propose ExpandingConv and CombConv layers which achieve state-of-the-art performance on a variety of graph tasks. We also show that multi-head GAT is one of the ExpandingConv implementations, which brings a theoretical explanation for its effectiveness.

## 2 PRELIMINARIES

### 2.1 NOTATIONS

For a graph $G(V, E)$, we denote the set of edges, nodes and node feature vectors respectively by $E_G$, $V_G$ and $X_G$. $\mathcal{N}(v)$ represents the set of neighbors of $v$ including itself, i.e., $\mathcal{N}(v) = \{u \in V_G | (u, v) \in E_G\} \cup \{v\}$. We use $[n]$ to denote the set $\{1, 2, ..., n\}$. $\{\{...\}\}$ represents a multi-set, i.e., a set with possibly repeating elements. $\Pi_n$ represents the set of all permutations of the integers 1 to $n$. $\boldsymbol{h}_\pi$, where $\pi \in \Pi_{|h|}$, is a reordering of the elements of a sequence $\boldsymbol{h}$ according to $\pi$. Given a matrix $\boldsymbol{X} \in \mathbb{R}^{a \times b}$, $\boldsymbol{X}^T$ represents the transpose of $\boldsymbol{X}$, and $\text{vec}(\boldsymbol{X}) \in \mathbb{R}^{ab \times 1}$ represents the column stack of $\boldsymbol{X}$.

### 2.2 GRAPH NEURAL NETWORKS

Most GNNs adopt the neighborhood aggregation scheme (Gilmer et al., 2017) to learn the node representations, which utilizes both node features and graph structures. In the $k$-th layer, the representation of node $v$

$$\boldsymbol{h}_v^{(k)} = \text{Update}(\boldsymbol{h}_v^{(k-1)}, \text{Aggregate}(\{\{\boldsymbol{h}_u^{(k-1)} | u \in \mathcal{N}(v)\}\})). \tag{1}$$

**Aggregators in GNNs.** An aggregator is a permutation invariant function (Zaheer et al., 2017) with bounded size inputs. It satisfies: (i) size insensitive: an aggregator can take an arbitrary but finite size of inputs; (ii) permutation invariant: an aggregator is invariant to the permutation of input. There are a limited number of basic aggregators such as SUM, MEAN, NORMALIZED MEAN, MAX/MIN, STD, etc. Most proposed GNNs apply one of these aggregators. Sum-of-power mapping (Zaheer et al., 2017) and normalized moments (Corso et al., 2020) can also be used as aggregators and they allow for a variable number of aggregators.

## 3 PROPOSED MODEL

In this section, we first formalize the distinguishing strength of aggregators as a partial order, and show why basic aggregators used in popular GNNs become bottlenecks of expressiveness. Then, we analyze the requirements for building powerful aggregators and even injective aggregators. Finally, we introduce two GNN layers based on our theoretical analysis.

### 3.1 Distinguishing Strength of Aggregators

To ensure generality, our analysis of aggregators is always considered in multisets and uncountable case, where the inputs are continuous and with possibly repeating elements. We first introduce distinguishing strength under the concept of partial order (Schmidt, 2011).

**Distinguishing strength.** The distinguishing strength of aggregator $\boldsymbol{f}_{aggr1}$ is stronger than $\boldsymbol{f}_{aggr2}$, denoted by $\boldsymbol{f}_{aggr1} \succeq \boldsymbol{f}_{aggr2}$, if and only if for any two multisets $\boldsymbol{x}_1$ and $\boldsymbol{x}_2$ where the number of elements can be different, $\boldsymbol{f}_{aggr2}(\boldsymbol{x}_1) \neq \boldsymbol{f}_{aggr2}(\boldsymbol{x}_2) \Rightarrow \boldsymbol{f}_{aggr1}(\boldsymbol{x}_1) \neq \boldsymbol{f}_{aggr1}(\boldsymbol{x}_2)$. Meanwhile, if there exist $\boldsymbol{x}_1'$ and $\boldsymbol{x}_2'$ such that $\boldsymbol{f}_{aggr1}(\boldsymbol{x}_1') \neq \boldsymbol{f}_{aggr1}(\boldsymbol{x}_2')$ but $\boldsymbol{f}_{aggr2}(\boldsymbol{x}_1') = \boldsymbol{f}_{aggr2}(\boldsymbol{x}_2')$, $\boldsymbol{f}_{aggr1}$ is strictly stronger than $\boldsymbol{f}_{aggr2}$, denoted by $\boldsymbol{f}_{aggr1} \succ \boldsymbol{f}_{aggr2}$. If $\boldsymbol{f}_{aggr1} \succeq \boldsymbol{f}_{aggr2} \succeq \boldsymbol{f}_{aggr1}$, we say these two aggregators have the same distinguishing strength, denoted by $\boldsymbol{f}_{aggr1} = \boldsymbol{f}_{aggr2}$. If there exist multisets $\boldsymbol{x}_1$ and $\boldsymbol{x}_2$ such that $\boldsymbol{f}_{aggr1}(\boldsymbol{x}_1) \neq \boldsymbol{f}_{aggr1}(\boldsymbol{x}_2)$, $\boldsymbol{f}_{aggr2}(\boldsymbol{x}_1) = \boldsymbol{f}_{aggr2}(\boldsymbol{x}_2)$, and there also exist $\boldsymbol{x}_1'$ and $\boldsymbol{x}_2'$ such that $\boldsymbol{f}_{aggr1}(\boldsymbol{x}_1') = \boldsymbol{f}_{aggr1}(\boldsymbol{x}_2')$, $\boldsymbol{f}_{aggr2}(\boldsymbol{x}_1') \neq \boldsymbol{f}_{aggr2}(\boldsymbol{x}_2')$, we say $\boldsymbol{f}_{aggr1}$ and $\boldsymbol{f}_{aggr2}$ are incomparable.

Distinguishing strength is a partial order, and the set of all aggregators form a poset. In this poset, the aggregators with the greatest distinguishing strength should be injective. With the definition of distinguishing strength, we can compare any two aggregators. The distinguishing strength of widely used aggregators SUM, MEAN, MAX/MIN is incomparable. One can easily give two multisets of elements that are distinguished by one aggregator but are not distinguished by the others as showed in (Corso et al., 2020).

**Equivariant aggregator.** $\boldsymbol{f}_{\text{aggr}} : \{\{\mathbb{R}^d\}\} \to \mathbb{R}^d$ is an equivariant aggregator if and only if $\boldsymbol{f}_{\text{aggr}}(\{\{\boldsymbol{T} \cdot \boldsymbol{x}_1, \boldsymbol{T} \cdot \boldsymbol{x}_2, \cdots, \boldsymbol{T} \cdot \boldsymbol{x}_n\}\}) = \boldsymbol{T} \cdot \boldsymbol{f}_{\text{aggr}}(\{\{\boldsymbol{x}_1, \boldsymbol{x}_2, \cdots, \boldsymbol{x}_n\}\})$ for any $\boldsymbol{T} \in \mathbb{R}^{m \times d}$ and $\{\{\boldsymbol{x}_i \in \mathbb{R}^d | i \in [n]\}\}$.

Widely used SUM and MEAN are equivariant aggregators but MAX/MIN is not. We denote $\boldsymbol{f}_{aggr1} \otimes \boldsymbol{f}_{aggr2}$ a new aggregator by combing $\boldsymbol{f}_{aggr1}$ and $\boldsymbol{f}_{aggr2}$ with $\boldsymbol{f}_{aggr1} \otimes \boldsymbol{f}_{aggr2}(X) = [\boldsymbol{f}_{aggr1}(X)||\boldsymbol{f}_{aggr2}(X)]$, where $||$ denotes concatenation.

**Lemma 1.** *(i) For any continuous function $g$, we have $g \circ \boldsymbol{f}_{aggr} \preceq \boldsymbol{f}_{aggr}$, and when $g$ is injective, $\boldsymbol{f}_{aggr} = g \circ \boldsymbol{f}_{aggr}$; (ii) $\boldsymbol{f}_{aggr1} \otimes \boldsymbol{f}_{aggr2} \succeq \boldsymbol{f}_{aggr1}$ and $\boldsymbol{f}_{aggr1} \otimes \boldsymbol{f}_{aggr2} \succeq \boldsymbol{f}_{aggr2}$. If $\boldsymbol{f}_{aggr1}$ and $\boldsymbol{f}_{aggr2}$ are incomparable, $\boldsymbol{f}_{aggr1} \otimes \boldsymbol{f}_{aggr2} \succ \boldsymbol{f}_{aggr1}$ and $\boldsymbol{f}_{aggr1} \otimes \boldsymbol{f}_{aggr2} \succ \boldsymbol{f}_{aggr2}$; (iii) If $\boldsymbol{f}_{aggr}$ is an equivariant aggregator, then $\boldsymbol{f}_{aggr}(\boldsymbol{T} \cdot \boldsymbol{x}_1, \boldsymbol{T} \cdot \boldsymbol{x}_2, \cdots, \boldsymbol{T} \cdot \boldsymbol{x}_n) \preceq \boldsymbol{f}_{aggr}(\boldsymbol{x}_1, \boldsymbol{x}_2, \cdots, \boldsymbol{x}_n)$ for any $\boldsymbol{T} \in \mathbb{R}^{m \times d}$ and $\{\{\boldsymbol{x}_i \in \mathbb{R}^d | i \in [n]\}\}$.*

We prove Lemma 1 in Appendix B. Lemma 1 indicates that aggregators become bottlenecks of distinguishing strength. For the equivariant aggregator, any linear transformation before aggregation and any transformation after aggregation have no contribution to the distinguishing strength. For SUM and MEAN, we have $g(\text{SUM}(\boldsymbol{T} \cdot \boldsymbol{x}_1, \boldsymbol{T} \cdot \boldsymbol{x}_2, \cdots, \boldsymbol{T} \cdot \boldsymbol{x}_n)) \preceq \text{SUM}(\boldsymbol{x}_1, \boldsymbol{x}_2, \cdots, \boldsymbol{x}_n)$ and $g(\text{MEAN}(\boldsymbol{T} \cdot \boldsymbol{x}_1, \boldsymbol{T} \cdot \boldsymbol{x}_2, \cdots, \boldsymbol{T} \cdot \boldsymbol{x}_n)) \preceq \text{MEAN}(\boldsymbol{x}_1, \boldsymbol{x}_2, \cdots, \boldsymbol{x}_n)$, where $\boldsymbol{T} \in \mathbb{R}^{m \times d}$, and $g$ can be any continuous function. Based on Lemma 1, we can now compare the distinguishing strength of aggregations in some popular GNNs. GIN-0 sums all hidden features of neighbors at first, and then pass them to a 2-layer MLP. Therefore, when considering in a continuous input features space, the distinguishing strength of GIN-0 is at most as powerful as the SUM aggregator. GCN uses a NORMALIZED MEAN (denoted by $n$MEAN) aggregator. Given a node $v$ and its neighbors, $n\text{MEAN}(v, u_1, \cdots, u_n) = \frac{1}{\sqrt{|\mathcal{N}(v)|}} \cdot (\frac{\boldsymbol{h}_v}{\sqrt{|\mathcal{N}(v)|}} + \frac{\boldsymbol{h}_{u_1}}{\sqrt{|\mathcal{N}(u_1)|}} + \cdots + \frac{\boldsymbol{h}_{u_n}}{\sqrt{|\mathcal{N}(u_n)|-1}})$. $n$MEAN is also an equivariant aggregator, and the distinguishing strength of aggregation in GCN is at most as powerful as $n$MEAN. GAT corresponds to the weighted SUM aggregation, where the weight coefficients are the functions of hidden features. This makes the distinguishing strength of GAT and SUM incomparable. Based on these observations, a potential approach to breaking the distinguishing strength limitation is to apply a nonlinear processing on inputs before aggregation.

### 3.2 Building Powerful Aggregators

In this section, we analyze the requirements for building more powerful aggregators and further injective aggregators. We first introduce a new representation of GNN layers which unifies several popular GNN layers. Given a node $v$ and its neighbors $\mathcal{N}(v)$, our new formulation represents the

GNN operation as follows:

$$
\begin{cases}
\boldsymbol{m}_v = \boldsymbol{f}_{\text{local}}(v) & \text{/* aggregation coefficients generation */} \\
\boldsymbol{r}_v^{(t)} = \boldsymbol{m}_{v\pi} \bar{\boldsymbol{h}}_{v\pi}^{(t-1)T} & \text{/* neighborhood aggregation */} \\
\boldsymbol{h}_v^{(t)} = \boldsymbol{f}_{\text{NN}}(\boldsymbol{r}_v^{(t)}) & \text{/* feature/structure extraction */.}
\end{cases}
\tag{2}
$$

Here, $\boldsymbol{m}_v \in \mathbb{R}^{|\mathcal{N}(v)|}$ is the aggregation coefficients vector of node $v$. Note that $\boldsymbol{m}_v$ should be the mapping of local structures such as node degrees, node or edge features of the $k$-hop neighbors assigned on node $v$ to ensure the same encoding of isomorphic graphs. $\bar{\boldsymbol{h}}_{v\pi}^{(t-1)T} = (\boldsymbol{h}_v^{(t-1)}, \boldsymbol{h}_{u_1}^{(t-1)}, \cdots, \boldsymbol{h}_{u_{|\mathcal{N}(v)|}}^{(t-1)})^T \in \mathbb{R}^{|\mathcal{N}(v)| \times d}$ is the matrix representation of $v$'s neighbors according to a permutation $\pi$. $\boldsymbol{f}_{\text{NN}} : \mathbb{R}^d \to \mathbb{R}^{d'}$ is a neural network that extracts task-relevant information from the aggregated representation $\boldsymbol{r}_v^{(t)}$, and is used to update hidden feature $\boldsymbol{h}_v^{(t)}$ of node $v$.

According to Equation 2, the aggregation should be with high distinguishing strength to avoid indistinguishability among neighbors. Meanwhile, the extraction should be powerful enough to efficiently extract task-relevant structural patterns from the aggregated representation of neighbors. Based on these observations, we reformulate GCN, GIN0 and GAT with their corresponding three-stage representations as follows:

$$
\text{GCN} : \begin{cases}
\boldsymbol{m}_v = \frac{1}{\sqrt{|\mathcal{N}(v)|}}\left(\frac{1}{\sqrt{|\mathcal{N}(v)|}}, \frac{1}{\sqrt{|\mathcal{N}(u_1)|}}, \cdots, \frac{1}{\sqrt{|\mathcal{N}(u_{|\mathcal{N}(v)|-1})|}}\right) \\
\boldsymbol{r}_v^{(t)} = \boldsymbol{m}_{v\pi} \bar{\boldsymbol{h}}_{v\pi}^{(t-1)T} \\
\boldsymbol{h}_v^{(t)} = \sigma(\boldsymbol{W}\boldsymbol{r}_v^{(t)} + \boldsymbol{b}),
\end{cases}
\quad
\text{GIN0} : \begin{cases}
\boldsymbol{m}_v = \boldsymbol{1}^{1 \times |\mathcal{N}(v)|} \\
\boldsymbol{r}_v^{(t)} = \boldsymbol{m}_{v\pi} \bar{\boldsymbol{h}}_{v\pi}^{(t-1)T} \\
\boldsymbol{h}_v^{(t)} = \text{MLP}(\boldsymbol{r}_v^{(t)}),
\end{cases}
$$

$$
\text{GAT} : \begin{cases}
\boldsymbol{m}_v^{(t)} = (\text{att}(\boldsymbol{h}_v^{(t-1)}, \boldsymbol{h}_v^{(t-1)}), \text{att}(\boldsymbol{h}_v^{(t-1)}, \boldsymbol{h}_{u_1}^{(t-1)}), \cdots, \text{att}(\boldsymbol{h}_v^{(t-1)}, \boldsymbol{h}_{u_{|\mathcal{N}(v)-1|}}^{(t-1)})) \\
\boldsymbol{r}_v^{(t)} = \boldsymbol{m}_{v\pi}^{(t)} \bar{\boldsymbol{h}}_{v\pi}^{(t-1)T} \\
\boldsymbol{h}_v^{(t)} = \sigma(\boldsymbol{W}\boldsymbol{r}_v^{(t)} + \boldsymbol{b}).
\end{cases}
$$

Their default formulations are given in Appendix A. In the aggregation step, GCN's $\boldsymbol{m}_v$ is the mapping of neighbors' degrees; GIN0's $\boldsymbol{m}_v$ is the mapping of node $v$'s degree which is equivalent to SUM aggregator; GAT's $\boldsymbol{m}_v$ is the mapping of neighbors' features. All of them are the mappings of local structures as given in Equation 2.

In this three-stage representation, the aggregation is reformulated as the multiplication of the aggregation coefficients vector and the feature matrix of neighbors. It provides insights on improving the distinguishing strength of aggregations. First, we show how to characterize the permutation invariance in this formulation. Let $\boldsymbol{M} \in \mathbb{R}^{s \times n}$ denote an aggregation coefficient matrix where $s \geqslant 1$. Note that in GCN, GIN and GAT, $s$ is restricted to be 1. $\boldsymbol{h}_\pi \in \mathbb{R}^{n \times d}$ is the matrix representation of $n$ input elements according to $\pi$. The aggregation computation in the second step of Equation 2 is $\boldsymbol{f}_{\text{aggr}}(\boldsymbol{M}, \boldsymbol{h}_\pi) = \boldsymbol{M}\boldsymbol{P}_\pi \boldsymbol{h}_\pi = \boldsymbol{M}_\pi \boldsymbol{h}_\pi$, where $\boldsymbol{P}_\pi$ is the permutation matrix according to $\pi$. $\boldsymbol{P}_\pi \boldsymbol{h}_\pi$ ensures the same output for all $\boldsymbol{h}_\pi$, $\pi \in \Pi_{|h|}$. $\boldsymbol{M}_\pi = \boldsymbol{M}\boldsymbol{P}_\pi$ is the reordering of columns of $\boldsymbol{M}$ according to $\pi$. For any $\pi_1, \pi_2 \in \Pi_n$, $\boldsymbol{f}_{\text{aggr}}(\boldsymbol{M}, \boldsymbol{h}_{\pi_1}) = \boldsymbol{f}_{\text{aggr}}(\boldsymbol{M}, \boldsymbol{h}_{\pi_2})$, thus permutation invariance holds. Once $\boldsymbol{M}$ is decided, we obtain a unique aggregator denoted by $\boldsymbol{f}_M$. For any sequence of input elements $\boldsymbol{h}$, $\boldsymbol{f}_M(\boldsymbol{h}) = \boldsymbol{f}_{\text{aggr}}(\boldsymbol{M}, \boldsymbol{h}_\pi)$, where $\pi \in \Pi_n$ can be any ordering of neighbors. Next, we analyze the distinguishing strength of $\boldsymbol{f}_M$.

**Proposition 1.** *For any two matrices $\boldsymbol{M} \in \mathbb{R}^{s \times n}$ and $\boldsymbol{M}' \in \mathbb{R}^{s' \times n}$ with $s, s' \leqslant n$, we have (i) $\boldsymbol{f}_{\left(\begin{smallmatrix} M \\ M' \end{smallmatrix}\right)} \geq \boldsymbol{f}_M$, where $\left(\begin{smallmatrix} M \\ M' \end{smallmatrix}\right)$ means stacking these two matrices; (ii) $\boldsymbol{f}_{\left(\begin{smallmatrix} M \\ M' \end{smallmatrix}\right)} > \boldsymbol{f}_M$ if and only if $rank(\left(\begin{smallmatrix} M \\ M' \end{smallmatrix}\right)) > rank(\boldsymbol{M})$; (iii) Any multiset of size $n$ is distinguishable with $\boldsymbol{f}_M$ if and only if $rank(\boldsymbol{M}) = n$.*

We prove Proposition 1 in Appendix C. Proposition 1 shows that the distinguishing strength of $\boldsymbol{f}_M$ is decided by the rank of the corresponding $\boldsymbol{M}$. Yet, the distinguishing strength analysis in Proposition 1 is only suitable for multisets aggregated with shared $\boldsymbol{f}_M$. Next, we extend the analysis for the case of different aggregators.

Let $\text{Res}(\boldsymbol{f}_M)$ denote the set of all outputs of $\boldsymbol{f}_M$. Our proposed three-stage representation also provides useful insight on the constraints among different aggregators. That is, in order to fully distinguish different local structures, for any two different $\boldsymbol{f}_{M_1}$ and $\boldsymbol{f}_{M_2}$, $\text{Res}(\boldsymbol{f}_{M_1}) \cap \text{Res}(\boldsymbol{f}_{M_2}) = \varnothing$. This is because to fully distinguish different local structures, we should ensure their aggregated

representations are different. Since $M$ is restricted to be the mapping of local structures (such as $k$-hop neighbors), different $M$ means that the corresponding local structures are different. Therefore, the aggregation results of different $f_M$ must be different. However, it is not satisfied by existing GNNs, and there are few studies on distinguishing multisets aggregated by different aggregators. In Proposition 2, we present a detailed analysis of it.

**Proposition 2.** *For any* $M_1, M_2 \in \mathbb{R}^{s \times n_1}$ *and* $M_1', M_2' \in \mathbb{R}^{s' \times n_2}$, *(i)* $Res(f_{\begin{pmatrix} M_1 \\ M_1' \end{pmatrix}}) \cap Res(f_{\begin{pmatrix} M_2 \\ M_2' \end{pmatrix}}) \subseteq Res(f_{M_1}) \cap Res(f_{M_2})$; *(ii) If* $Res(f_{\begin{pmatrix} M_1 \\ M_1' \end{pmatrix}}) \cap Res(f_{\begin{pmatrix} M_2 \\ M_2' \end{pmatrix}}) \subset Res(f_{M_1}) \cap Res(f_{M_2})$, *then* $rank(\begin{pmatrix} M_1 & M_2 \\ M_1' & M_2' \end{pmatrix}) > rank(\begin{pmatrix} M_1 & M_2 \end{pmatrix})$;

We prove Proposition 2 in Appendix D. Proposition 2 shows the necessity of preserving the rank of aggregation coefficient matrix when considering the distinguishing strength among different aggregators. Next, we provide a sufficient condition for building desired multiple injective aggregators with the outputs having no intersections.

**Proposition 3.** *For any two aggregators* $f_{M_1}$ *and* $f_{M_2}$ *with* $M_1 \in \mathbb{R}^{s \times n_1}$ *and* $M_2 \in \mathbb{R}^{s \times n_2}$, *if* $rank(\begin{pmatrix} M_1 & M_2 \end{pmatrix}) = n_1 + n_2$, *then* $f_{M_1}$ *and* $f_{M_2}$ *are injective and* $Res(f_{M_1}) \cap Res(f_{M_2}) = \varnothing$.

We prove Proposition 3 in Appendix E. Proposition 1, 2 and 3 provide a new perspective for building powerful aggregators and even injective aggregators. Compared with the distinguishing strength studies in (Xu et al., 2019) and (Corso et al., 2020), as well as existing strategies for building injective aggregators, e.g., sum-of-power mapping (Zaheer et al., 2017) and normalized moments (Corso et al., 2020), we reformulate the aggregation with aggregation coefficients matrix and show the relations of the distinguishing strength of aggregators and the rank of the corresponding aggregation coefficients matrices. Besides, the aggregation of this method is controlled by aggregation coefficients which can be learned from graph data to better leverage structural information. In this paper, to simplify the analysis, we only consider the aggregations within one-hop neighbors. The results can be easily extended to more sophisticated aggregators with the overall framework unchanged

In the perspective of preserving the rank of hidden features among neighbors, $r = Mh$ indicates that $\text{rank}(r) \leqslant \min(\text{rank}(M), \text{rank}(h))$. To preserve the rank of hidden features in aggregations such that $\text{rank}(r) = \text{rank}(h)$, we need $\text{rank}(M) \geqslant \text{rank}(h)$. This builds a connection between improving the distinguishing strength of aggregators and preserving the rank of hidden features among neighbors, both of which have the requirements on the rank of $M$. General aggregators such as ones in GCN, GIN-0 and GAT have $\text{rank}(M) = 1$. Thus, $\text{rank}(r)$ is always fixed to 1 no matter what the rank of the input features is. Correspondingly, they have a weak distinguishing strength.

Equation 2 splits the aggregation and feature/structure extraction into two independent steps, which helps to figure out that the expressive power loss happens in the aggregation step, and then the model extracts feature/structure information on the distorted encodings of neighbors. From Equation 2, the aggregation can be considered as a representation regularization step, which unifies different multisets of neighbors into the same representation style while holding permutation invariance. Then, the model can extract structural information on this regulated data with a shared trainable matrix as the third step in Equation 2. Based on this observation, we propose two novel GNN layers: ExpandingConv and CombConv.

## 3.3 EXPANDINGCONV

In this section, we first present ExpandingConv framework. Then we provide one of its implementations and analyze how ExpandingConv achieves more powerful aggregations. The ExpandingConv framework is

$$\begin{cases} m_{uv}^{(t)} = f_{\text{local}}(u, v)|_{u \in \mathcal{N}(v)} \\ h_v^{(t)} = f_{\text{aggr}}(\{\{\text{vec}(m_{uv}^{(t)} h_u^{(t-1)T})|u \in \mathcal{N}(v)\}\}), \end{cases}$$

where $m_{uv}^{(t)} \in \mathbb{R}^{s \times 1}$ with $s > 1$ and $f_{\text{local}}(u, v)$ is the mapping of local structures between nodes $u$ and $v$. The implementation of $f_{\text{local}}(u, v)$ is very flexible with the only restriction of ensuring the same encoding of isomorphic graphs. $\text{vec}(m_{uv}^{(t)} h_u^{(t-1)T}) \in \mathbb{R}^{sd \times 1}$ is the expanded representation of hidden features $h_u^{(t-1)} \in \mathbb{R}^{d \times 1}$. Then a GNN layer $f_{\text{aggr}}$ learns structural information on this

expanded representations. We introduce an implelentation as follows:

$$
\begin{cases}
\boldsymbol{m}_{uv}^{(t)} = \text{Tanh}(\boldsymbol{W}[\boldsymbol{h}_v^{(t-1)}||\boldsymbol{h}_u^{(t-1)}] + \boldsymbol{b}) \\
\boldsymbol{h}_v^{(t)} = \sum_{u\in\mathcal{N}(v)} \text{MLP}(\text{vec}(\boldsymbol{m}_{uv}^{(t)}\boldsymbol{h}_u^{(t-1)T})).
\end{cases}
\tag{3}
$$

In Equation 3, we implement $\boldsymbol{f}_{\text{local}}(u,v)$ as the function of hidden features of nodes $u$ and $v$. There can be other implementations, and we leave them for future work. $\boldsymbol{W} \in \mathbb{R}^{s\times 2d}$ and $\boldsymbol{b} \in \mathbb{R}^{s\times 1}$ are trainable matrices. (Luan et al., 2019) empirically showed that different nonlinear activatoin functions have different contributions in preserving the rank of matrices. We use the recommended Tanh as the activation function in the computation of $\boldsymbol{m}_{uv}^{(t)}$ to better preserve the rank of aggregation coefficient matrices. MLP denotes a 2-layer perceptron.

Next, we represent Equation 3 with the corresponding three-stage representation as given in Section 3.2 to obtain its aggregation coefficient matrix and analyze its distinguishing strength. To simplify this process, we only consider 1-layer MLP with $\boldsymbol{W}' \in \mathbb{R}^{d\times sd}$ and $\boldsymbol{b}' \in \mathbb{R}^{d\times 1}$.

$$
\boldsymbol{h}_v^{(t)} = \sum_{u\in\mathcal{N}(v)} \text{ReLU}(\boldsymbol{W}'\text{vec}(\boldsymbol{m}_{uv}^{(t)}\boldsymbol{h}_u^{(t-1)T}) + \boldsymbol{b}') =
$$

$$
\begin{pmatrix}
\sum_{u\in\mathcal{N}(v)} \text{ReLU}(\boldsymbol{W}'_{[1,:]}\text{vec}(\boldsymbol{m}_{uv}^{(t)}\boldsymbol{h}_u^{(t-1)T}) + \boldsymbol{b}'_{[1]}) \\
\sum_{u\in\mathcal{N}(v)} \text{ReLU}(\boldsymbol{W}'_{[2,:]}\text{vec}(\boldsymbol{m}_{uv}^{(t)}\boldsymbol{h}_u^{(t-1)T}) + \boldsymbol{b}'_{[2]}) \\
\vdots \\
\sum_{u\in\mathcal{N}(v)} \text{ReLU}(\boldsymbol{W}'_{[d,:]}\text{vec}(\boldsymbol{m}_{uv}^{(t)}\boldsymbol{h}_u^{(t-1)T}) + \boldsymbol{b}'_{[d]})
\end{pmatrix}
=
\begin{pmatrix}
\sum_{u\in\mathcal{N}_1(v)} (\boldsymbol{W}'_{[1,:]}\text{vec}(\boldsymbol{m}_{uv}^{(t)}\boldsymbol{h}_u^{(t-1)T}) + \boldsymbol{b}'_{[1]}) \\
\sum_{u\in\mathcal{N}_2(v)} (\boldsymbol{W}'_{[2,:]}\text{vec}(\boldsymbol{m}_{uv}^{(t)}\boldsymbol{h}_u^{(t-1)T}) + \boldsymbol{b}'_{[2]}) \\
\vdots \\
\sum_{u\in\mathcal{N}_d(v)} (\boldsymbol{W}'_{[d,:]}\text{vec}(\boldsymbol{m}_{uv}^{(t)}\boldsymbol{h}_u^{(t-1)T}) + \boldsymbol{b}'_{[d]})
\end{pmatrix}
$$

$$
=
\begin{pmatrix}
\boldsymbol{W}'_{[1,:]} & & & \\
& \boldsymbol{W}'_{[2,:]} & & \\
& & \ddots & \\
& & & \boldsymbol{W}'_{[d,:]}
\end{pmatrix}
\begin{pmatrix}
\text{vec}(\sum_{u\in\mathcal{N}_1(v)} \boldsymbol{m}_{uv}^{(t)}\boldsymbol{h}_u^{(t-1)T}) \\
\text{vec}(\sum_{u\in\mathcal{N}_2(v)} \boldsymbol{m}_{uv}^{(t)}\boldsymbol{h}_u^{(t-1)T}) \\
\vdots \\
\text{vec}(\sum_{u\in\mathcal{N}_d(v)} \boldsymbol{m}_{uv}^{(t)}\boldsymbol{h}_u^{(t-1)T})
\end{pmatrix}
+
\begin{pmatrix}
|\mathcal{N}_1(v)| \cdot \boldsymbol{b}'_{[1]} \\
|\mathcal{N}_2(v)| \cdot \boldsymbol{b}'_{[2]} \\
\vdots \\
|\mathcal{N}_d(v)| \cdot \boldsymbol{b}'_{[d]}
\end{pmatrix}
$$

$$
=
\begin{pmatrix}
\boldsymbol{W}'_{[1,:]} & & & \\
& \boldsymbol{W}'_{[2,:]} & & \\
& & \ddots & \\
& & & \boldsymbol{W}'_{[d,:]}
\end{pmatrix}
\begin{pmatrix}
\text{vec}(\boldsymbol{M}_{v_1\pi_1}^{(t)}\bar{\boldsymbol{h}}_{v_1\pi_1}^{(t-1)T}) \\
\text{vec}(\boldsymbol{M}_{v_2\pi_2}^{(t)}\bar{\boldsymbol{h}}_{v_2\pi_2}^{(t-1)T}) \\
\vdots \\
\text{vec}(\boldsymbol{M}_{v_d\pi_d}^{(t)}\bar{\boldsymbol{h}}_{v_d\pi_d}^{(t-1)T})
\end{pmatrix}
+
\begin{pmatrix}
|\mathcal{N}_1(v)| \cdot \boldsymbol{b}'_{[1]} \\
|\mathcal{N}_2(v)| \cdot \boldsymbol{b}'_{[2]} \\
\vdots \\
|\mathcal{N}_d(v)| \cdot \boldsymbol{b}'_{[d]}
\end{pmatrix},
\tag{4}
$$

where $\mathcal{N}_i(v) \subseteq \mathcal{N}(v)|i \in [d]$ are sampled subsets of neighbors in each dimension. $\boldsymbol{M}_{v_i}^{(t)} = (\boldsymbol{m}_{u_1v}^{(t)}, \boldsymbol{m}_{u_2v}^{(t)}, \cdots, \boldsymbol{m}_{u_{|\mathcal{N}_i(v)|}v}^{(t)}) \in \mathbb{R}^{s\times|\mathcal{N}_i(v)|}$ and $\bar{\boldsymbol{h}}_{v_i}^{(t-1)} = (\boldsymbol{h}_{u_1}^{(t-1)}, \boldsymbol{h}_{u_2}^{(t-1)}, \cdots, \boldsymbol{h}_{u_{|\mathcal{N}_i(v)|}}^{(t-1)}) \in \mathbb{R}^{d\times|\mathcal{N}_i(v)|}$ are aggregation coefficients matrix and hidden feature matrix corresponding to the subset of neighbors $\mathcal{N}_i(v) \subseteq \mathcal{N}(v)$ according to $\pi_i$. We denote $[\boldsymbol{h}_v^{(t-1)}||\bar{\boldsymbol{h}}_{v_i}^{(t-1)}] = (\boldsymbol{h}_v^{(t-1)}||\boldsymbol{h}_{u_1}^{(t-1)}, \boldsymbol{h}_v^{(t-1)}||\boldsymbol{h}_{u_2}^{(t-1)}, \cdots, \boldsymbol{h}_v^{(t-1)}||\boldsymbol{h}_{u_{|\mathcal{N}_i(v)|}}^{(t-1)}) \in \mathbb{R}^{2d\times|\mathcal{N}_i(v)|}$, then $\boldsymbol{M}_{v_i}^{(t)} = \text{Tanh}(\boldsymbol{W}[\boldsymbol{h}_v^{(t-1)}||\bar{\boldsymbol{h}}_{v_i}^{(t-1)}] + \boldsymbol{b}) \in \mathbb{R}^{s\times|\mathcal{N}_i(v)|}$. According to Equation 4, we finally obtain the three-stage representation equivalent to Equation 3.

$$
\begin{cases}
\boldsymbol{M}_{v_i}^{(t)} = \text{Tanh}(\boldsymbol{W}[\boldsymbol{h}_v^{(t-1)}||\bar{\boldsymbol{h}}_{v_i}^{(t-1)}] + \boldsymbol{b}) \\
\bar{\boldsymbol{r}}_{v_i}^{(t)} = \boldsymbol{M}_{v_i\pi_i}^{(t)}\bar{\boldsymbol{h}}_{v_i\pi_i}^{(t-1)T}, \quad i \in [d] \\
\boldsymbol{h}_v^{(t)} = \text{diag}(\boldsymbol{W}'_{[1,:]}, \boldsymbol{W}'_{[2,:]}, \cdots, \boldsymbol{W}'_{[d,:]})(\text{vec}(\bar{\boldsymbol{r}}_{v_1}^{(t)}), \text{vec}(\bar{\boldsymbol{r}}_{v_2}^{(t)}), \cdots, \text{vec}(\bar{\boldsymbol{r}}_{v_d}^{(t)}))^T \\
\quad\quad +(|\mathcal{N}_1(v)| \cdot \boldsymbol{b}'_{[1]}, |\mathcal{N}_2(v)| \cdot \boldsymbol{b}'_{[2]}, \cdots, |\mathcal{N}_d(v)| \cdot \boldsymbol{b}'_{[d]})^T.
\end{cases}
\tag{5}
$$

According to the computation of $\bar{\boldsymbol{r}}_{v_i}^{(t)}$, $\text{rank}(\bar{\boldsymbol{r}}_{v_i}^{(t)}) \leqslant \min(s, d, |\mathcal{N}_i(v)|)$. By configuring a larger $s$, we have $\text{rank}(\bar{\boldsymbol{r}}_{v_i}^{(t)}) > 1$ with a high probability, which is different from general GNNs with a rank of 1. As analyzed in Section 3.2, this achieves more powerful aggregators as well as preserves the rank of hidden features among neighbors. The obtained $\bar{\boldsymbol{r}}_{v_i}^{(t)}$ after aggregation is the unified representations of neighbors. We then use the trainable matrix $\boldsymbol{W}' \in \mathbb{R}^{sd\times d}$ to extracts feature/structure information. Unlike the aggregation step, the dimensions reduction here (from $sd$ to $d$) would not cause information loss. This can be explained by the fact that only task-relevant structural information needs to be preserved and passed to the next layer, and it can be embedded in lower dimensions.

**Comparisons with multi-head GAT.**

**Proposition 4.** *Multi-head GAT is an implementation of ExpandingConv as follows:*

$$
\begin{cases}
\boldsymbol{\alpha}_{vu} = softmax(LeakyReLU([diag(\tilde{\boldsymbol{a}}^{1T}, \tilde{\boldsymbol{a}}^{2T}, \cdots, \tilde{\boldsymbol{a}}^{KT}) \\
\quad ||diag(\tilde{\boldsymbol{a}}'^{1T}, \tilde{\boldsymbol{a}}'^{2T}, \cdots, \tilde{\boldsymbol{a}}'^{KT})][\boldsymbol{W}\boldsymbol{h}_v^{(t-1)}||\boldsymbol{W}\boldsymbol{h}_u^{(t-1)}])) \\
\boldsymbol{h}_v^{(t)} = \sigma\left(\frac{1}{K}\boldsymbol{W}\sum_{j\in\mathcal{N}_v} vec(\boldsymbol{\alpha}_{vu}\boldsymbol{h}_u^{(t-1)T})\right),
\end{cases}
\tag{6}
$$

*where $\boldsymbol{W} = ||_{k=1}^K \boldsymbol{W}^k \in \mathbb{R}^{kd\times d}$ is the concatenation of the trainable matrix in all $K$ heads.*

We prove Proposition 4 in Appendix F. Although multi-head GAT is based on attention mechanism, ExpandingConv provides a new perspective to explain its effectiveness. Applying multi-head attention mechanism helps to preserve the rank of hidden features as well as achieve more powerful aggregators. However, the usage of LeakyReLU may be harmful to preserving the rank of the aggregation coefficient matrix (Luan et al., 2019).

GAT as well as most other GNNs (such as GCN, GIN, etc) follows the same pattern that applies nonlinear units after aggregation. According to the analysis in Section 3.1, Equation 3 applies MLP on $vec(\boldsymbol{m}_{uv}^{(t)}\boldsymbol{h}_u^{(t-1)T})$ before SUM to break the distinguishing strength limitation of SUM. It also produces other interesting results. By reformulating Equation 3 with its three-stage representation as Equation 5, each dimension of hidden features aggregates on a subset of neighbors independently, which corresponds to a kind of dimension-wise neighbor sampling mechanism. We call the modification of applying ReLU ahead of SUM aggregator as $Re$-SUM mechanism. (Mishra et al., 2020) and (Rong et al., 2019) studied dropedge and node masking mechanism on node-level predictions, both of which can be considered as neighbor sampling strategies that have shown their effectiveness in improving the generalization ability of aggregation-based GNNs and are also used as unbiased data augmentation technique for training. Compared with dropedge and node masking, $Re$-SUM realizes a dimension-wise neighbor sampling, and it does not need to manually set the sampling ratio since this mechanism takes effects implicitly. $Re$-SUM shows that the neural network itself can perform sampling by properly combining nonlinear units and aggregators, without explicitly modifying the network architecture. Our experimental results verified the effectiveness of the $Re$-SUM on a variety of graph tasks.

## 3.4 COMBCONV

The CombConv framework is

$$
\begin{cases}
\boldsymbol{m}_{uv}^{(t)} = \boldsymbol{f}_{\text{local}}(u, v)|_{u\in\mathcal{N}(v)} \\
\boldsymbol{h}_v^{(t)} = \boldsymbol{f}_{\text{aggr}}(\{\{vec(\boldsymbol{m}_{uv}^{(t)}\odot\boldsymbol{h}_u^{(t-1)})|u\in\mathcal{N}(v)\}\}),
\end{cases}
$$

where $\boldsymbol{m}_{uv}^{(t)} \in \mathbb{R}^{d\times 1}$ and $\odot$ denotes element-wise product. An implementation of CombConv is given as follows:

$$
\begin{cases}
\boldsymbol{m}_{uv}^{(t)} = \text{Tanh}(\boldsymbol{W}[\boldsymbol{h}_v^{(t-1)}||\boldsymbol{h}_u^{(t-1)}] + \boldsymbol{b}) \\
\boldsymbol{h}_v^{(t)} = \sum_{u\in\mathcal{N}(v)} \text{MLP}(\boldsymbol{m}_{uv}^{(t)}\odot\boldsymbol{h}_u^{(t-1)}),
\end{cases}
\tag{7}
$$

where $\boldsymbol{W} \in \mathbb{R}^{d\times 2d}$ and $\boldsymbol{b} \in \mathbb{R}^{d\times 1}$. Similar to ExpandingConv, CombConv also applies $Re$-SUM aggregation. The difference is that each dimension of hidden features is aggregated with an independent weighted aggregator. ExpandingConv with $s = 1$ corresponds to a special case of CombConv where all dimensions share the same aggregator. Therefore, the distinguishing strength of CombConv is stronger than ExpandingConv with $s = 1$. Meanwhile, CombConv does not expand the hidden features of nodes in aggregation. Hence, it requires fewer parameters.

## 4 EXPERIMENTS

In this section, we evaluate ExpandingConv and CombConv on graph-level prediction tasks on OGB (Weihua Hu, 2020), TU (Kersting et al., 2016; Yanardag & Vishwanathan, 2015) and QM9 (Ramakrishnan et al., 2014; Wu et al., 2018; Ruddigkeit et al., 2012). The code is available at `https://github.com/qslim/epcb-gnns`.

**Configurations.** We use the default dataset splits for OGB. The QM9 dataset is randomly split into 80% train, 10% validation and 10% test as given in (Morris et al., 2019; Maron et al., 2019). For TU dataset, we follow the standard 10-fold cross validation protocol and splits from (Zhang et al., 2018) and report our results following the protocol described in (Xu et al., 2019; Ying et al., 2018). We use the concatenation of hidden features from all layers to compute the entire graph representations (Xu et al., 2018). In our tests, all models are equipped with batch normalization (Ioffe & Szegedy, 2015) on each hidden layer when evaluating on OGB and TU, and are not when evaluating on QM9. All datasets' descriptions and detailed hyperparameter settings are given in Appendix H.

We first conduct comprehensive ablation studies to evaluate the effectiveness of powerful aggregators and $Re$-SUM mechanism on OGB and QM9 as given in Table 1 and Table 2. Then, we compare the performance of ExpandingConv and CombConv with competitive baselines on all three datasets as given in Table 3 and Table 4 to show their improvements. ExpC-$s$ denotes ExpandingConv with $W \in \mathbb{R}^{s \times \star}$. We use ExpC* and CombC* to denote the ExpandingConv and CombConv without $Re$-SUM.

Table 1: Ablation studies on OGB and QM9. Higher is better.

|  | ogbg-ppa | ogbg-molhiv | ogbg-molpcba | ogbg-code |
| --- | --- | --- | --- | --- |
| ExpC*-1 | 70.65 | 77.63 | 22.65 | 32.2 |
| ExpC-1 | 77.50 | 76.79 | 23.39 | 32.6 |
| ExpC-3,4,5 | **80.11** | **77.89** | 23.44 | **33.2** |
| CombC* | 73.61 | 76.47 | 23.45 | 32.29 |
| CombC | 77.64 | 76.63 | **23.73** | 32.72 |

Table 2: Ablation studies on QM9. Lower is better.

|  | $\mu$ | $\alpha$ | $\epsilon_{homo}$ | $\epsilon_{lumo}$ | $\Delta\epsilon$ | $\langle R^2 \rangle$ | $ZPVE$ | $U_0$ | $U$ | $H$ | $G$ | $C_v$ |
| --- | --- | --- | --- | --- | --- | --- | --- | --- | --- | --- | --- | --- |
| ExpC*-1 | 0.467 | 0.283 | 0.00337 | 0.00340 | 0.00467 | 22.9 | 0.000205 | 0.0255 | 0.0263 | 0.0242 | 0.0261 | 0.1189 |
| ExpC-1 | 0.469 | 0.268 | 0.00326 | 0.00329 | 0.00466 | 20.8 | 0.000186 | 0.0202 | 0.0199 | 0.0202 | 0.0201 | 0.1039 |
| ExpC-4 | 0.413 | 0.255 | 0.00273 | 0.00300 | 0.00420 | 19.4 | **0.000168** | 0.0184 | 0.0183 | 0.0178 | 0.0182 | 0.1115 |
| ExpC-8 | 0.400 | 0.257 | 0.00259 | 0.00286 | 0.00395 | 18.1 | 0.000172 | 0.0158 | 0.0170 | 0.0177 | 0.0184 | 0.1060 |
| ExpC-16 | **0.382** | 0.255 | **0.00248** | **0.00268** | **0.00373** | 17.2 | 0.000170 | 0.0170 | 0.0174 | 0.0193 | **0.0165** | 0.1043 |
| ExpC-32 | **0.368** | 0.244 | **0.00248** | **0.00257** | **0.00364** | **16.3** | 0.000174 | **0.0151** | **0.0167** | **0.0165** | 0.0198 | **0.0962** |
| CombC* | 0.4062 | 0.248 | 0.00259 | 0.00273 | 0.00387 | 17.1 | 0.000170 | 0.0185 | 0.0181 | 0.0164 | 0.0174 | 0.1022 |
| CombC | 0.399 | **0.241** | 0.00261 | 0.00278 | 0.00386 | **15.9** | **0.000160** | **0.0144** | **0.0145** | **0.0147** | **0.0140** | **0.0858** |

## 4.1 ABLATION STUDIES

**Effect of powerful aggregators.** For complex graph structures with dense connections or with abundant node/edge features, they would benefit from a higher expressive model to maximumly distinguish different structures and extract relevant structural patterns as the model goes deeper to leverage large receptive fields. This is validated on both QM9 and OGB. We configure $s = 1, 4, 8, 16, 32$ of ExpC-$s$ for all 12 targets of QM9. As we apply a larger $s$, the model continuously achieves better performance on most targets. We randomly select $s = 1, 4$ for ogbg-ppa, ogbg-molhiv and $s = 1, 5$ for ogbg-code. The results show that applying larger $s$ gains performance improvements, especially on ogbg-ppa which involves large graphs with dense connections.

**Effect of $Re$-SUM mechanism.** In Table 1 and Table 2, the performance differences between ExpC*-1 (CombC*) and ExpC-1 (CombC) show the effectiveness of $Re$-SUM. In our tests, the $Re$-SUM can be extremely powerful on graphs with dense connections such as ogbg-ppa, which is validated on both ExpandingConv (with 6.85% improvements) and CombConv (with 4% improvements). On most targets of QM9, this mechanism also gains improvements. For small graphs with sparse connections such as ogbg-hiv and ogbg-molpcba, the improvements are not very significant.

## 4.2 COMPARISONS WITH BASELINES

Table 3 and Table 4 show the performance comparisons of our models with baselines on QM9, TU and OGB respectively. All datasets in QM9 and OGB graph-level predictions are used for evaluations. For TU, we use 3 widely used datasets: COLLAB includes graphs with dense connections;

Table 3: Comparisons with baselines on OGB and TU. Higher is better.

| | OGB | | | | TU | | |
|---|---|---|---|---|---|---|---|
| | ogbg-ppa | ogbg-molhiv | ogbg-molpcba | ogbg-code | COLLAB | RDT-B | RDT-M12 |
| DGK (Yanardag & Vishwanathan, 2015) | NA | NA | NA | NA | $73.09 \pm 0.25$ | $78.04 \pm 0.39$ | $32.22 \pm 0.1$ |
| PSCN (Niepert et al., 2016) | NA | NA | NA | NA | $73.76 \pm 0.50$ | $86.30 \pm 1.58$ | $41.32 \pm 0.42$ |
| AWE (Ivanov & Burnaev, 2018) | NA | NA | NA | NA | $73.93 \pm 1.94$ | $87.89 \pm 2.53$ | $39.20 \pm 2.09$ |
| GCN (Kipf & Welling, 2016) | $68.39 \pm 0.84$ | $76.06 \pm 0.97$ | $20.20 \pm 0.24$ | $31.63 \pm 0.18$ | NA | NA | NA |
| GIN (Xu et al., 2019) | $68.92 \pm 1.0$ | $75.58 \pm 1.40$ | $22.66 \pm 0.28$ | $31.63 \pm 0.20$ | $80.2 \pm 1.9$ | $92.4 \pm 2.5$ | NA |
| GraphSAG(Hamilton et al., 2017) | NA | NA | NA | NA | 68.25 | NA | 42.24 |
| DiffPool (Ying et al., 2018) | NA | NA | NA | NA | 75.48 | NA | 47.08 |
| CapsGNN (Xinyi & Chen, 2019) | NA | NA | NA | NA | $79.62 \pm 0.91$ | NA | $46.62 \pm 1.9$ |
| PPGN (Maron et al., 2019) | NA | NA | NA | NA | $80.16 \pm 1.1$ | NA | NA |
| DeeperGCN (Li et al., 2020a) | $77.12 \pm 0.71$ | $78.58 \pm 1.17$ | NA | NA | NA | NA | NA |
| HIMP (Fey et al., 2020) | NA | $\mathbf{78.80 \pm 0.82}$ | NA | NA | NA | NA | NA |
| WEGL (Kolouri et al., 2020) | NA | $77.57 \pm 1.11$ | NA | NA | NA | NA | NA |
| multi-head GAT(Veličković et al., 2017) | NA | 75.81 | 20.10 | 31.10 | NA | NA | NA |
| ExpC-$s$ | $\mathbf{79.76 \pm 0.72}$ | $77.99 \pm 0.82$ | $23.42 \pm 0.29$ | $\mathbf{33.18 \pm 0.17}$ | $\mathbf{82.10 \pm 1.60}$ | $92.2 \pm 1.87$ | $\mathbf{49.91 \pm 1.75}$ |
| CombC | $77.81 \pm 0.76$ | $77.15 \pm 1.32$ | $\mathbf{23.63 \pm 0.23}$ | $32.76 \pm 0.15$ | $81.90 \pm 1.75$ | $\mathbf{92.5 \pm 1.69}$ | $49.02 \pm 1.21$ |

Table 4: Comparisons with baselines on QM9. Lower is better.

| | $\mu$ | $\alpha$ | $\epsilon_{homo}$ | $\epsilon_{lumo}$ | $\Delta\epsilon$ | $\langle R^2 \rangle$ | $ZPVE$ | $U_0$ | $U$ | $H$ | $G$ | $C_v$ |
|---|---|---|---|---|---|---|---|---|---|---|---|---|
| DTNN (Wu et al., 2018) | **0.244** | 0.95 | 0.00388 | 0.00512 | 0.0112 | 17 | 0.00172 | 2.43 | 2.43 | 2.43 | 2.43 | 0.27 |
| MPNN (Gilmer et al., 2017) | 0.358 | 0.89 | 0.00541 | 0.00623 | 0.0066 | 28.5 | 0.00216 | 2.05 | 2 | 2.02 | 2.02 | 0.42 |
| k-GNN (Morris et al., 2019) | 0.476 | 0.27 | 0.00337 | 0.00351 | 0.0048 | 22.9 | 0.00019 | 0.0427 | 0.111 | 0.0419 | 0.0469 | **0.0944** |
| PPGN (Maron et al., 2019) | **0.0934** | 0.318 | **0.00174** | **0.0021** | **0.0029** | **3.78** | 0.000399 | 0.022 | 0.0504 | 0.0294 | 0.024 | 0.144 |
| GIN0* (Xu et al., 2019) | 0.471 | 0.281 | 0.00327 | 0.00340 | 0.00473 | 22.9 | 0.000202 | 0.0244 | 0.0245 | 0.0233 | 0.0255 | 0.1283 |
| GAT-$s$(Veličković et al., 2017) | 0.452 | 0.286 | 0.00322 | 0.00327 | 0.00460 | 22.7 | 0.000228 | 0.0212 | 0.0223 | 0.0223 | 0.0219 | 0.1247 |
| ExpC-$s$ | 0.368 | **0.244** | **0.00248** | **0.00257** | **0.00364** | 16.3 | **0.000168** | **0.0151** | **0.0167** | **0.0165** | **0.0165** | 0.0962 |
| CombC | 0.399 | **0.241** | 0.00261 | 0.00278 | 0.00386 | **15.9** | **0.000160** | **0.0144** | **0.0145** | **0.0147** | **0.0140** | **0.0858** |

REDDIT-BINARY (RDT-B) and REDDIT-MULTI-12K (RDT-M12) include large and sparse graphs with one center node having dense connections with other nodes. All results of baselines are taken from the original papers except for the results of GraphSAGE on TU, multi-head GAT on OGB and GIN0* on QM9 which were not reported by the original papers. We report the results of Graph-SAGE provided by (Ying et al., 2018) and evaluate multi-head GAT and GIN0* by ourselves. To ensure a fair comparison, for OGB and TU, we configure the number of heads in multi-head GAT and $s$ in ExpC-$s$ to be the same which is selected in $\{3, 4, 5\}$. For QM9, the number of heads is 8 and $s \in \{4, 8, 16, 32\}$. GIN0* in QM9 denotes GIN0 without batch normalization.

Compared with baselines, our models achieve the best performance on 7 out of all 12 targets of QM9, 3 out of all 4 graph-level prediction datasets of OGB and all 3 selected TU datasets. Our models get 1.9% improvements on COLLAB and 2.83% improvements on REDDIT-MULTI-12K compared with SOTA baselines. On ogbg-ppa, our models achieve 2.6% higher classification accuracies compared with SOTA baselines. On ogbg-code, they achieve 1.5% improvements. Multi-head GAT can also be considered as an implementation of ExpandingConv. However, its performance on graph-level predictions is not competitive. According to its three-stage representation, the usage of LeakyReLU in the aggregation step is harmful to preserving the rank, and the usage of softmax makes it harder to analyze the rank. In the extraction step, the 1-layer MLP may have a limited representation power to represent the desired extraction functions.

## 5 CONCLUSION

We show how basic aggregators used in general GNNs become expressive bottlenecks. To address this limitation, we develop theoretical foundations of building powerful aggregators. We also propose the $Re$-SUM mechanism which achieves dimension-wise sampling. To evaluate their effectiveness, we develop two novel GNN layers, and conduct extensive experiments on public graph benchmarks. The results are consistent with our analysis, and our proposed models achieve SOTA performance on a variety of graph-level prediction benchmarks.

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

## A    GCN, GAT AND GIN

Here, we present implementations of GCN, GAT and GIN for the usage of our analysis.

**Graph Convolution Networks (GCN) (Kipf & Welling, 2016).**

$$H^{(t)} = \sigma(\hat{D}^{-\frac{1}{2}}\hat{A}\hat{D}^{-\frac{1}{2}}H^{(t-1)}W + b) \tag{8}$$

**Graph Attention Networks (GAT) (Veličković et al., 2017).**

$$h_v^{(t)} = \sigma\left(\sum_{u \in \mathcal{N}(v)} \text{att}\left(h_v^{(t-1)}, h_u^{(t-1)}\right) W h_u^{(t-1)}\right) \tag{9}$$

**Graph Isomorphism Networks (GIN-0) (Xu et al., 2019).**

$$h_v^{(t)} = \text{MLP}(\sum_{u \in \mathcal{N}(v)} h_u^{(t-1)}). \tag{10}$$

## B    PROOF OF LEMMA 1

*Proof.* (i) For any two multisets $\boldsymbol{x}_1$ and $\boldsymbol{x}_2$, if $g(\boldsymbol{f}_{\text{aggr}}(\boldsymbol{x}_1)) \neq g(\boldsymbol{f}_{\text{aggr}}(\boldsymbol{x}_2))$, then $\boldsymbol{f}_{\text{aggr}}(\boldsymbol{x}_1) \neq \boldsymbol{f}_{\text{aggr}}(\boldsymbol{x}_2)$. Therefore, we have $\boldsymbol{f}_{\text{aggr}} \succeq g \circ \boldsymbol{f}_{\text{aggr}}$. If $g$ is injective, then $\boldsymbol{f}_{\text{aggr}}(\boldsymbol{x}_1) \neq \boldsymbol{f}_{\text{aggr}}(\boldsymbol{x}_2) \Rightarrow g(\boldsymbol{f}_{\text{aggr}}(\boldsymbol{x}_1)) \neq g(\boldsymbol{f}_{\text{aggr}}(\boldsymbol{x}_2))$. We have $\boldsymbol{f}_{\text{aggr}} \succeq g \circ \boldsymbol{f}_{\text{aggr}} \succeq \boldsymbol{f}_{\text{aggr}}$, therefore $\boldsymbol{f}_{\text{aggr}} = g \circ \boldsymbol{f}_{\text{aggr}}$.

(ii) For any two multisets $\boldsymbol{x}_1$ and $\boldsymbol{x}_2$, $\boldsymbol{f}_{aggr1}(\boldsymbol{x}_1) \neq \boldsymbol{f}_{aggr1}(\boldsymbol{x}_2) \Rightarrow [\boldsymbol{f}_{aggr1}(\boldsymbol{x}_1)\|\boldsymbol{f}_{aggr2}(\boldsymbol{x}_1)] \neq [\boldsymbol{f}_{aggr1}(\boldsymbol{x}_2)\|\boldsymbol{f}_{aggr2}(\boldsymbol{x}_2)]$. Therefore, $\boldsymbol{f}_{aggr1} \otimes \boldsymbol{f}_{aggr2} \succeq \boldsymbol{f}_{aggr1}$. If $\boldsymbol{f}_{aggr1}$ and $\boldsymbol{f}_{aggr2}$ are incomparable, there exist $\boldsymbol{x}_3$ and $\boldsymbol{x}_4$ such that $\boldsymbol{f}_{aggr2}(\boldsymbol{x}_3) \neq \boldsymbol{f}_{aggr2}(\boldsymbol{x}_4)$ but $\boldsymbol{f}_{aggr1}(\boldsymbol{x}_3) = \boldsymbol{f}_{aggr1}(\boldsymbol{x}_4)$. Therefore, there exist $\boldsymbol{x}_3$ and $\boldsymbol{x}_4$ such that $[\boldsymbol{f}_{aggr1}(\boldsymbol{x}_3)\|\boldsymbol{f}_{aggr2}(\boldsymbol{x}_3)] \neq [\boldsymbol{f}_{aggr1}(\boldsymbol{x}_4)\|\boldsymbol{f}_{aggr2}(\boldsymbol{x}_4)]$ but $\boldsymbol{f}_{aggr1}(\boldsymbol{x}_3) = \boldsymbol{f}_{aggr1}(\boldsymbol{x}_4)$. $\boldsymbol{f}_{aggr1} \otimes \boldsymbol{f}_{aggr2} \succ \boldsymbol{f}_{aggr1}$.

(iii) Since $\boldsymbol{f}_{\text{aggr}}$ is an equivariant aggregator, then $\boldsymbol{f}_{\text{aggr}}(\boldsymbol{T} \cdot \boldsymbol{x}_1, \boldsymbol{T} \cdot \boldsymbol{x}_2, \cdots, \boldsymbol{T} \cdot \boldsymbol{x}_n) = \boldsymbol{T} \cdot \boldsymbol{f}_{\text{aggr}}(\boldsymbol{x}_1, \boldsymbol{x}_2, \cdots, \boldsymbol{x}_n) \preceq \boldsymbol{f}_{\text{aggr}}(\boldsymbol{x}_1, \boldsymbol{x}_2, \cdots, \boldsymbol{x}_n)$.    □

## C    PROOF OF PROPOSITION 1

*Proof.* (i)
$$\boldsymbol{f}_{\left(\begin{smallmatrix} M \\ M' \end{smallmatrix}\right)}(\boldsymbol{x}) = \left(\begin{smallmatrix} M \\ M' \end{smallmatrix}\right)_\pi \boldsymbol{x}_\pi = \begin{pmatrix} \boldsymbol{m}_\pi \boldsymbol{x}_\pi \\ \boldsymbol{m}'_\pi \boldsymbol{x}_\pi \end{pmatrix} = \begin{pmatrix} \boldsymbol{f}_M(\boldsymbol{x}) \\ \boldsymbol{f}_{M'}(\boldsymbol{x}) \end{pmatrix},$$

then for any $\boldsymbol{x}_1$ and $\boldsymbol{x}_2$, we have

$$\boldsymbol{f}_M(\boldsymbol{x}_1) \neq \boldsymbol{f}_M(\boldsymbol{x}_2) \Rightarrow \boldsymbol{f}_{\left(\begin{smallmatrix} M \\ M' \end{smallmatrix}\right)}(\boldsymbol{x}_1) \neq \boldsymbol{f}_{\left(\begin{smallmatrix} M \\ M' \end{smallmatrix}\right)}(\boldsymbol{x}_2),$$

and therefore we conclude that $\boldsymbol{f}_{\left(\begin{smallmatrix} M \\ M' \end{smallmatrix}\right)} \succeq \boldsymbol{f}_M$.

(ii) "$\boldsymbol{f}_{\left(\begin{smallmatrix} M \\ M' \end{smallmatrix}\right)} \succ \boldsymbol{f}_M \Leftarrow \text{rank}(\left(\begin{smallmatrix} M \\ M' \end{smallmatrix}\right)) > \text{rank}(M)$"

We prove the claim by contradiction. Assume that $\text{rank}(\left(\begin{smallmatrix} M \\ M' \end{smallmatrix}\right)) > \text{rank}(M)$ and $\boldsymbol{f}_{\left(\begin{smallmatrix} M \\ M' \end{smallmatrix}\right)} = \boldsymbol{f}_M$. $\boldsymbol{f}_{\left(\begin{smallmatrix} M \\ M' \end{smallmatrix}\right)} = \boldsymbol{f}_M$ means that for any $\boldsymbol{x}_1$ and $\boldsymbol{x}_2$, $\boldsymbol{m}_\pi \boldsymbol{x}_{1\pi} = \boldsymbol{m}_\pi \boldsymbol{x}_{2\pi} \Leftrightarrow \left(\begin{smallmatrix} M \\ M' \end{smallmatrix}\right)_\pi \boldsymbol{x}_{1\pi} = \left(\begin{smallmatrix} M \\ M' \end{smallmatrix}\right)_\pi \boldsymbol{x}_{2\pi}$, where $\pi$ is the ordering of input elements. Let $\boldsymbol{s} = \boldsymbol{x}_{1\pi} - \boldsymbol{x}_{2\pi}$. For any $i \in [n]$, $\boldsymbol{s}[i] = \boldsymbol{x}_{1\pi}[i] - \boldsymbol{x}_{2\pi}[i] \in \mathbb{R}$. Then for any $\boldsymbol{s} \in \mathbb{R}^n$, $\boldsymbol{m}_\pi \boldsymbol{s} = \boldsymbol{0} \Leftrightarrow \left(\begin{smallmatrix} M \\ M' \end{smallmatrix}\right)_\pi \boldsymbol{s} = \boldsymbol{0}$. The system of linear equations $\boldsymbol{m}_\pi \boldsymbol{x} = \boldsymbol{0}$ and $\left(\begin{smallmatrix} M \\ M' \end{smallmatrix}\right)_\pi \boldsymbol{x} = \boldsymbol{0}$ share the same solution space. Let $R_S$ denote the rank of this solution space, then $\text{rank}(\boldsymbol{m}_\pi) + R_S = \text{rank}(\left(\begin{smallmatrix} M \\ M' \end{smallmatrix}\right)_\pi) + R_S = n$. Therefore, $\text{rank}(\boldsymbol{m}_\pi) = \text{rank}(\left(\begin{smallmatrix} M \\ M' \end{smallmatrix}\right)_\pi)$, then we have $\text{rank}(M) = \text{rank}(\left(\begin{smallmatrix} M \\ M' \end{smallmatrix}\right))$. Since we assumed that $\text{rank}(\left(\begin{smallmatrix} M \\ M' \end{smallmatrix}\right)) > \text{rank}(M)$, we reach a contradiction.

"$\boldsymbol{f}_{\left(\begin{smallmatrix} M \\ M' \end{smallmatrix}\right)} \succ \boldsymbol{f}_M \Rightarrow \text{rank}(\left(\begin{smallmatrix} M \\ M' \end{smallmatrix}\right)) > \text{rank}(M)$"

We prove an equivalent proposition "$\text{rank}(\left(\begin{smallmatrix} M \\ M' \end{smallmatrix}\right)) \leqslant \text{rank}(M) \Rightarrow f_{\left(\begin{smallmatrix} M \\ M' \end{smallmatrix}\right)} \preceq f_M$". Note that $\text{rank}(\left(\begin{smallmatrix} M \\ M' \end{smallmatrix}\right)) \geqslant \text{rank}(M)$ and $f_{\left(\begin{smallmatrix} M \\ M' \end{smallmatrix}\right)} \succeq f_M$ as given in Proposition 1(i). We only need to prove "$\text{rank}(\left(\begin{smallmatrix} M \\ M' \end{smallmatrix}\right)) = \text{rank}(M) \Rightarrow f_{\left(\begin{smallmatrix} M \\ M' \end{smallmatrix}\right)} = f_M$". $\text{rank}(\left(\begin{smallmatrix} M \\ M' \end{smallmatrix}\right)) = \text{rank}(M)$ means that any row in $M'$ is linearly dependent to rows in $M$. Therefore, there exists $L \in \mathbb{R}^{s' \times s}$ so that $\left(\begin{smallmatrix} M \\ M' \end{smallmatrix}\right) = \left(\begin{smallmatrix} I \\ L \end{smallmatrix}\right) M$. For any $x_1$ and $x_2$ with $M P_\pi x_{1\pi} = M P_\pi x_{2\pi}$, $\left(\begin{smallmatrix} I \\ L \end{smallmatrix}\right) M P_\pi x_{1\pi} = \left(\begin{smallmatrix} I \\ L \end{smallmatrix}\right) M P_\pi x_{2\pi}$, and therefore $\left(\begin{smallmatrix} M \\ M' \end{smallmatrix}\right) P_\pi x_{1\pi} = \left(\begin{smallmatrix} M \\ M' \end{smallmatrix}\right) P_\pi x_{2\pi}$, where $\pi$ is the ordering of input elements. That is, for any $x_1$ and $x_2$, $f_M(x_1) = f_M(x_2) \Rightarrow f_{\left(\begin{smallmatrix} M \\ M' \end{smallmatrix}\right)}(x_1) = f_{\left(\begin{smallmatrix} M \\ M' \end{smallmatrix}\right)}(x_2)$, thus $f_{\left(\begin{smallmatrix} M \\ M' \end{smallmatrix}\right)} \preceq f_M$. Finally, we have $f_{\left(\begin{smallmatrix} M \\ M' \end{smallmatrix}\right)} = f_M$.

(iii) "Any multiset of size $n$ is distinguishable with $f_M \Rightarrow \text{rank}(M) = n$"

Since $\text{rank}(M) \leqslant n$, we prove an equivalent proposition "$\text{rank}(M) < n \Rightarrow$ there exists at least two multisets which are indistinguishable". Considering the system of linear equations $y = Mx$ where $x \in \mathbb{R}^n$, if $\text{rank}(M) < n$, then there exists $y'$ such that $\text{rank}(M) = \text{rank}(M, y') < n$. According to the Rouché–Capelli theorem, there are infinite solutions $x_i'$ such that $y' = Mx_1' = Mx_2' = \cdots = Mx_\infty'$, Each $x_i'$ comes from a multiset with a particular order. Next, we need to prove that all these $x_i'$ come from more than one multiset. As a multiset with bounded size $n$ constitutes at most $n!$ different orders, the infinite number of $x_i'$ corresponds to $y'$ must come from more than one multisets, making these multisets indistinguishable.

"Any multiset of size $n$ is distinguishable with $f_M \Leftarrow \text{rank}(M) = n$"

Since $\text{rank}(M) = n$ and $s = n$, for any $x \in \mathbb{R}^n$, $y = Mx \in \mathbb{R}^n$ is unique. Correspondingly, for any $P_\pi x_\pi$, $M(P_\pi x_\pi)$ is unique.

$\square$

# D    PROOF OF PROPOSITION 2

*Proof.* (i) According to the proof of Proposition 1(i), $f_{\left(\begin{smallmatrix} M \\ M' \end{smallmatrix}\right)}(x) = \begin{pmatrix} f_M(x) \\ f_{M'}(x) \end{pmatrix}$. For any $x_1$ and $x_2$ with $f_{\left(\begin{smallmatrix} M_1 \\ M_1' \end{smallmatrix}\right)}(x_1) = f_{\left(\begin{smallmatrix} M_2 \\ M_2' \end{smallmatrix}\right)}(x_2)$, $f_{M_1}(x_1) = f_{M_2}(x_2)$ holds. Meanwhile, $f_{\left(\begin{smallmatrix} M_1 \\ M_1' \end{smallmatrix}\right)}(x_1) = f_{\left(\begin{smallmatrix} M_2 \\ M_2' \end{smallmatrix}\right)}(x_2) \in \text{Res}(f_{\left(\begin{smallmatrix} M_1 \\ M_1' \end{smallmatrix}\right)}) \cap \text{Res}(f_{\left(\begin{smallmatrix} M_2 \\ M_2' \end{smallmatrix}\right)})$, and $f_{M_1}(x_1) = f_{M_2}(x_2) \in \text{Res}(f_{M_1}) \cap \text{Res}(f_{M_2})$. Therefore, for any $e \in \text{Res}(f_{\left(\begin{smallmatrix} M_1 \\ M_1' \end{smallmatrix}\right)}) \cap \text{Res}(f_{\left(\begin{smallmatrix} M_2 \\ M_2' \end{smallmatrix}\right)})$, we have $e \in \text{Res}(f_{M_1}) \cap \text{Res}(f_{M_2})$. That is $\text{Res}(f_{\left(\begin{smallmatrix} M_1 \\ M_1' \end{smallmatrix}\right)}) \cap \text{Res}(f_{\left(\begin{smallmatrix} M_2 \\ M_2' \end{smallmatrix}\right)}) \subseteq \text{Res}(f_{M_1}) \cap \text{Res}(f_{M_2})$.

(ii) We prove an equivalent proposition "$\text{Res}(f_{\left(\begin{smallmatrix} M_1 \\ M_1' \end{smallmatrix}\right)}) \cap \text{Res}(f_{\left(\begin{smallmatrix} M_2 \\ M_2' \end{smallmatrix}\right)}) = \text{Res}(f_{M_1}) \cap \text{Res}(f_{M_2}) \Leftarrow \text{rank}(\left(\begin{smallmatrix} M_1 & M_2 \\ M_1' & M_2' \end{smallmatrix}\right)) = \text{rank}(\left(\begin{smallmatrix} M_1 & M_2 \end{smallmatrix}\right))$". $\text{rank}(\left(\begin{smallmatrix} M_1 & M_2 \\ M_1' & M_2' \end{smallmatrix}\right)) = \text{rank}(\left(\begin{smallmatrix} M_1 & M_2 \end{smallmatrix}\right))$ means that any row in $\left(\begin{smallmatrix} M_1' & M_2' \end{smallmatrix}\right)$ is linearly dependent to $\left(\begin{smallmatrix} M_1 & M_2 \end{smallmatrix}\right)$. Therefore, there exists $L \in \mathbb{R}^{s' \times s}$ so that $\left(\begin{smallmatrix} M_1 & M_2 \\ M_1' & M_2' \end{smallmatrix}\right) = \left(\begin{smallmatrix} I \\ L \end{smallmatrix}\right)\left(\begin{smallmatrix} M_1 & M_2 \end{smallmatrix}\right)$. Correspondingly, $\left(\begin{smallmatrix} M_1 \\ M_1' \end{smallmatrix}\right) = \left(\begin{smallmatrix} I \\ L \end{smallmatrix}\right) M_1$ and $\left(\begin{smallmatrix} M_2 \\ M_2' \end{smallmatrix}\right) = \left(\begin{smallmatrix} I \\ L \end{smallmatrix}\right) M_2$. For any $x_1$ and $x_2$ with $M_1 P_\pi x_{1\pi} = M_2 P_\pi x_{2\pi}$, $\left(\begin{smallmatrix} I \\ L \end{smallmatrix}\right) M_1 P_\pi x_{1\pi} = \left(\begin{smallmatrix} I \\ L \end{smallmatrix}\right) M_2 P_\pi x_{2\pi}$, and therefore $\left(\begin{smallmatrix} M_1 \\ M_1' \end{smallmatrix}\right) P_\pi x_{1\pi} = \left(\begin{smallmatrix} M_2 \\ M_2' \end{smallmatrix}\right) P_\pi x_{2\pi}$, where $\pi$ is the ordering of input elements. Thus for any $x_1$ and $x_2$, $f_{M_1}(x_1) = f_{M_2}(x_2) \in \text{Res}(f_{M_1}) \cap \text{Res}(f_{M_2}) \Rightarrow f_{\left(\begin{smallmatrix} M_1 \\ M_1' \end{smallmatrix}\right)}(x_1) = f_{\left(\begin{smallmatrix} M_2 \\ M_2' \end{smallmatrix}\right)}(x_2) \in \text{Res}(f_{\left(\begin{smallmatrix} M_1 \\ M_1' \end{smallmatrix}\right)}) \cap \text{Res}(f_{\left(\begin{smallmatrix} M_2 \\ M_2' \end{smallmatrix}\right)})$. Hence, $\text{Res}(f_{M_1}) \cap \text{Res}(f_{M_2}) \subseteq \text{Res}(f_{\left(\begin{smallmatrix} M_1 \\ M_1' \end{smallmatrix}\right)}) \cap \text{Res}(f_{\left(\begin{smallmatrix} M_2 \\ M_2' \end{smallmatrix}\right)})$. According to Proposition 2(i), $\text{Res}(f_{M_1}) \cap \text{Res}(f_{M_2}) = \text{Res}(f_{\left(\begin{smallmatrix} M_1 \\ M_1' \end{smallmatrix}\right)}) \cap \text{Res}(f_{\left(\begin{smallmatrix} M_2 \\ M_2' \end{smallmatrix}\right)})$.

$\square$

## E  PROOF OF PROPOSITION 3

*Proof.* Since $\boldsymbol{M}_1 \in \mathbb{R}^{s \times n_1}$, $\boldsymbol{M}_2 \in \mathbb{R}^{s \times n_2}$ and $\text{rank}(\begin{pmatrix} \boldsymbol{M}_1 & \boldsymbol{M}_2 \end{pmatrix}) = n_1 + n_2$, we have $\text{rank}(\boldsymbol{M}_1) = n_1$ and $\text{rank}(\boldsymbol{M}_2) = n_2$. According to Proposition 1, $\boldsymbol{f}_{\boldsymbol{M}_1}$ and $\boldsymbol{f}_{\boldsymbol{M}_2}$ are injective.

We build the system of linear equations $\boldsymbol{y} = \boldsymbol{A}\boldsymbol{x}$, where $\boldsymbol{x} \in \mathbb{R}^{n_1+n_2}$, and $\boldsymbol{A} = \begin{pmatrix} \boldsymbol{M}_1 & \boldsymbol{M}_2 \end{pmatrix} \begin{pmatrix} \boldsymbol{I}^{n_1} & \boldsymbol{0} \\ \boldsymbol{0} & -\boldsymbol{I}^{n_2} \end{pmatrix} \in \mathbb{R}^{s \times (n_1+n_2)}$. Then, $\text{rank}(\boldsymbol{A}) = \text{rank}(\begin{pmatrix} \boldsymbol{M}_1 & \boldsymbol{M}_2 \end{pmatrix} \begin{pmatrix} \boldsymbol{I}^{n_1} & \boldsymbol{0} \\ \boldsymbol{0} & -\boldsymbol{I}^{n_2} \end{pmatrix}) = \text{rank}(\begin{pmatrix} \boldsymbol{M}_1 & \boldsymbol{M}_2 \end{pmatrix}) = n_1 + n_2$, which means $\boldsymbol{A}\boldsymbol{x} = \boldsymbol{0}$ has no non-zero solutions. Let $\boldsymbol{x}' = (\boldsymbol{x}[1], \boldsymbol{x}[2], \cdots, \boldsymbol{x}[n_1])$ and $\boldsymbol{x}'' = (\boldsymbol{x}[n_1 + 1], \boldsymbol{x}[n_1 + 2], \cdots, \boldsymbol{x}[n_1 + n_2])$ such that $\boldsymbol{x} = \begin{pmatrix} \boldsymbol{x}' \\ \boldsymbol{x}'' \end{pmatrix}$. For any $\boldsymbol{x} \neq \boldsymbol{0}$,

$$\boldsymbol{A}\boldsymbol{x} = \begin{pmatrix} \boldsymbol{M}_1 & \boldsymbol{M}_2 \end{pmatrix} \begin{pmatrix} \boldsymbol{I}^{n_1} & \boldsymbol{0} \\ \boldsymbol{0} & -\boldsymbol{I}^{n_2} \end{pmatrix} \begin{pmatrix} \boldsymbol{x}' \\ \boldsymbol{x}'' \end{pmatrix} = \boldsymbol{M}_1\boldsymbol{x}' - \boldsymbol{M}_2\boldsymbol{x}'' \neq \boldsymbol{0}.$$

Therefore, for any $\boldsymbol{x}' \in \mathbb{R}^{n_1}$, $\boldsymbol{x}'' \in \mathbb{R}^{n_2}$ and $\boldsymbol{x}', \boldsymbol{x}'' \neq \boldsymbol{0}$, $\boldsymbol{M}_1\boldsymbol{x}' \neq \boldsymbol{M}_2\boldsymbol{x}''$, hence $\boldsymbol{M}_1(\boldsymbol{P}_\pi \boldsymbol{x}'_\pi) \neq \boldsymbol{M}_2(\boldsymbol{P}_\pi \boldsymbol{x}''_\pi)$ for any $\boldsymbol{P}_\pi \boldsymbol{x}'_\pi$ and $\boldsymbol{P}_\pi \boldsymbol{x}''_\pi$. As a result, $\text{Res}(\boldsymbol{f}_{\boldsymbol{M}_1}) \cap \text{Res}(\boldsymbol{f}_{\boldsymbol{M}_2}) = \varnothing$. □

## F  PROOF OF PROPOSITION 4

*Proof.* For Multi-head GAT, there are two types of implementations on aggregating each head, *concatenation* and *average*. Here, we only consider the average aggregation implementation.

$$
\begin{aligned}
\boldsymbol{h}_v^{(t)} &= \sigma\left(\frac{1}{K}\sum_{k=1}^{K}\sum_{j\in\mathcal{N}_v}\alpha_{vu}^k \boldsymbol{W}^k \boldsymbol{h}_u^{(t-1)}\right) \\
&= \sigma\left(\frac{1}{K}\sum_{j\in\mathcal{N}_v}\sum_{k=1}^{K}\boldsymbol{W}^k(\alpha_{vu}^k \boldsymbol{h}_u^{(t-1)})\right) \\
&= \sigma\left(\frac{1}{K}\sum_{j\in\mathcal{N}_v}(\|_{k=1}^{K}\boldsymbol{W}^k)(\|_{k=1}^{K}\alpha_{vu}^k \boldsymbol{h}_u^{(t-1)})\right) \\
&= \sigma\left(\frac{1}{K}\sum_{j\in\mathcal{N}_v}\boldsymbol{W}\text{vec}(\boldsymbol{\alpha}_{vu}\boldsymbol{h}_u^{(t-1)T})\right) \\
&= \sigma\left(\frac{1}{K}\boldsymbol{W}\sum_{j\in\mathcal{N}_v}\text{vec}(\boldsymbol{\alpha}_{vu}\boldsymbol{h}_u^{(t-1)T})\right),
\end{aligned}
$$

where $\boldsymbol{W}^k \in \mathbb{R}^{d \times d}$ is the trainable matrix for the $k$-th head, and $\boldsymbol{W} = \|_{k=1}^{K}\boldsymbol{W}^k \in \mathbb{R}^{kd \times d}$ is the concatenation of the trainable matrix in all $K$ heads;

$$
\boldsymbol{\alpha}_{vu} = \text{softmax}(\text{LeakyReLU}\begin{pmatrix} \boldsymbol{a}^{1T}[\boldsymbol{W}^1\boldsymbol{h}_v^{(t-1)}\|\boldsymbol{W}^1\boldsymbol{h}_u^{(t-1)}] \\ \boldsymbol{a}^{2T}[\boldsymbol{W}^2\boldsymbol{h}_v^{(t-1)}\|\boldsymbol{W}^2\boldsymbol{h}_u^{(t-1)}] \\ \vdots \\ \boldsymbol{a}^{KT}[\boldsymbol{W}^1\boldsymbol{h}_v^{(t-1)}\|\boldsymbol{W}^K\boldsymbol{h}_u^{(t-1)}] \end{pmatrix}).
$$

Let $\tilde{a}^{\star} = (a_1^{\star}, a_2^{\star}, \cdots, a_K^{\star})$ and $\tilde{a}'^{\star} = (a_{K+1}^{\star}, a_{K+2}^{\star}, \cdots, a_{2K}^{\star})$. Then,

$$
\begin{pmatrix}
a^{1T}[W^1 h_v^{(t-1)} || W^1 h_u^{(t-1)}] \\
a^{2T}[W^2 h_v^{(t-1)} || W^2 h_u^{(t-1)}] \\
\vdots \\
a^{KT}[W^1 h_v^{(t-1)} || W^K h_u^{(t-1)}]
\end{pmatrix}
$$

$$
= \begin{pmatrix}
[\tilde{a}^1 || \tilde{a}'^1]^T [W^1 h_v^{(t-1)} || W^1 h_u^{(t-1)}] \\
[\tilde{a}^2 || \tilde{a}'^2]^T [W^2 h_v^{(t-1)} || W^2 h_u^{(t-1)}] \\
\vdots \\
[\tilde{a}^K || \tilde{a}'^K]^T [W^1 h_v^{(t-1)} || W^K h_u^{(t-1)}]
\end{pmatrix}
$$

$$
= \begin{pmatrix}
\tilde{a}^{1T} W^1 h_v^{(t-1)} + \tilde{a}'^{1T} W^1 h_u^{(t-1)} \\
\tilde{a}^{2T} W^2 h_v^{(t-1)} + \tilde{a}'^{2T} W^2 h_u^{(t-1)} \\
\vdots \\
\tilde{a}^{KT} W^K h_v^{(t-1)} + \tilde{a}'^{KT} W^K h_u^{(t-1)}
\end{pmatrix}
$$

$$
= \begin{pmatrix}
\tilde{a}^{1T} W^1 h_v^{(t-1)} \\
\tilde{a}^{2T} W^2 h_v^{(t-1)} \\
\vdots \\
\tilde{a}^{KT} W^K h_v^{(t-1)}
\end{pmatrix}
+ \begin{pmatrix}
\tilde{a}'^{1T} W^1 h_u^{(t-1)} \\
\tilde{a}'^{2T} W^2 h_u^{(t-1)} \\
\vdots \\
\tilde{a}'^{KT} W^K h_u^{(t-1)}
\end{pmatrix}
$$

$$
= \begin{pmatrix}
\tilde{a}^{1T} & & & \\
& \tilde{a}^{2T} & & \\
& & \ddots & \\
& & & \tilde{a}^{KT}
\end{pmatrix}
\begin{pmatrix}
W^1 \\ W^2 \\ \vdots \\ W^K
\end{pmatrix} h_v^{(t-1)}
+ \begin{pmatrix}
\tilde{a}'^{1T} & & & \\
& \tilde{a}'^{2T} & & \\
& & \ddots & \\
& & & \tilde{a}'^{KT}
\end{pmatrix}
\begin{pmatrix}
W^1 \\ W^2 \\ \vdots \\ W^K
\end{pmatrix} h_u^{(t-1)}
$$

$$
= \mathrm{diag}(\tilde{a}^{1T}, \tilde{a}^{2T}, \cdots, \tilde{a}^{KT}) W h_v^{(t-1)} + \mathrm{diag}(\tilde{a}'^{1T}, \tilde{a}'^{2T}, \cdots, \tilde{a}'^{KT}) W h_u^{(t-1)}
$$

$$
= [\mathrm{diag}(\tilde{a}^{1T}, \tilde{a}^{2T}, \cdots, \tilde{a}^{KT}) || \mathrm{diag}(\tilde{a}'^{1T}, \tilde{a}'^{2T}, \cdots, \tilde{a}'^{KT})][W h_v^{(t-1)} || W h_u^{(t-1)}].
$$

Therefore, multi-head GAT is an implementation of ExpandingConv as follows:

$$
\begin{cases}
\boldsymbol{\alpha}_{vu} = \mathrm{softmax}(\mathrm{LeakyReLU}([\mathrm{diag}(\tilde{a}^{1T}, \tilde{a}^{2T}, \cdots, \tilde{a}^{KT}) \\
\qquad\qquad || \mathrm{diag}(\tilde{a}'^{1T}, \tilde{a}'^{2T}, \cdots, \tilde{a}'^{KT})][W h_v^{(t-1)} || W h_u^{(t-1)}])) \\
h_v^{(t)} = \sigma\left( \frac{1}{K} W \sum_{j \in \mathcal{N}_v} \mathrm{vec}(\boldsymbol{\alpha}_{vu} h_u^{(t-1)T}) \right).
\end{cases}
$$

$\square$

## G    COMPARISONS WITH MULTI-AGGREGATOR IMPLEMENTATIONS

ExpandingConv can also be considered as a kind of multi-aggregator scheme. In Equation 5, each row of $M_{v_i}$ can be viewed as a weighted aggregator where the weight coefficients are learned from data. Proposition 1 shows that to obtain higher distinguishing strength by utilizing more aggregators, the weight coefficients of newly added aggregators should be linearly independent to all existing aggregators. The distinguishing strength of weighted aggregators is incomparable with basic aggregators. However, since each row of $M_{v_i}$ is equivalent to an independent aggregator, one can simply modify the implementation of $f_{\mathrm{local}}(u, v)$ to obtain the variant whose distinguishing strength is strict stronger than basic aggregators as follows:

$$
\mathrm{ExpandingConv}\big|_{m_{uv}'^{(t)} = [m_{uv}^{(t)} || 1]} > \mathrm{SUM},
$$

$$
\mathrm{ExpandingConv}\big|_{m_{uv}'^{(t)} = [m_{uv}^{(t)} || 1 || \frac{1}{|\mathcal{N}(v)|}]} > \mathrm{SUM} \otimes \mathrm{MEAN}.
$$

Compared with lerveraging multiple basic aggregators in (Corso et al., 2020) and (Dehmamy et al., 2019), lerveraging weighted aggregator allows for variable numbers of aggregators. Meanwhile, the weighted coefficients are learned from data, which can better capture relevant structural patterns.

## H    DETAILS OF EXPERIMENTAL SETUP

**Datasets.**   Benchmark datasets for graph kernels provided by TU (Kersting et al., 2016) suffer from their small scales of data, making them not sufficient to evaluate the performance of models (Dwivedi et al., 2020). Our evaluations are conducted on graph property predictions datasets ogbg-ppa, ogbg-code, ogbg-molhiv in OGB (Weihua Hu, 2020) and QM9 (Ramakrishnan et al., 2014; Wu et al., 2018; Ruddigkeit et al., 2012) which are large-scale graph datasets including graph classification and graph regression tasks. ogbg-ppa is extracted from the protein-protein association networks with large and densely connected graphs. ogbg-code is a collection of Abstract Syntax Trees (ASTs) obtained from Python method definitions with large and sparse graphs. ogbg-molhiv is molecular property prediction datasets with relative small graphs. QM9 consists 134K small organic molecules with the task to predict 12 targets for each molecule. All data is obtained from pytorch-geometric library (Fey & Lenssen, 2019).

Table 5: Hyperparameter settings for OGB.

| | ogbg-ppa | | ogbg-molhiv | | ogbg-molpcba | | ogbg-code | |
|---|---|---|---|---|---|---|---|---|
| | ExpC*-1, ExpC-$s$ | CombC*, CombC | ExpC*-1, ExpC-$s$ | CombC*, CombC | ExpC*-1, ExpC-$s$ | CombC*, CombC | ExpC*-1, ExpC-$s$ | CombC*, CombC |
| batch size | 32 | 32 | 64 | 64 | 128 | 128 | 64 | 64 |
| layers | 4 | 4 | 3 | 3 | 5 | 5 | 4 | 4 |
| hidden | 256 | 256 | 64 | 64 | 512 | 512 | 512 | 512 |
| lr | 0.0005 | 0.0002 | 0.0001 | 0.0001 | 0.0001 | 0.0001 | 0.0001 | 0.0001 |
| step size | 20 | 20 | 5 | 5 | 10 | 10 | 5 | 5 |
| lr decay | 0.8 | 0.7 | 0.7 | 0.7 | 0.6 | 0.6 | 0.6 | 0.6 |
| dropout | 0.5 | 0.5 | 0.5 | 0.5 | 0.5 | 0.5 | 0.5 | 0.5 |
| readout | SUM | SUM | MEAN | MEAN | MEAN | MEAN | MEAN | MEAN |

Table 6: Hyperparameter settings for TU.

| | COLLAB | | REDDIT-BINARY | | REDDIT-MULTI-12K | |
|---|---|---|---|---|---|---|
| | ExpC-$s$ | CombC | ExpC-$s$ | CombC | ExpC-$s$ | CombC |
| batch size | 32 | 32 | 64 | 64 | 64 | 64 |
| layers | 3 | 3 | 3 | 3 | 3 | 3 |
| hidden | 180 | 180 | 256 | 256 | 256 | 256 |
| lr | 0.001 | 0.001 | 0.001 | 0.001 | 0.001 | 0.001 |
| step size | 10 | 10 | 10 | 10 | 10 | 10 |
| lr decay | 0.8 | 0.8 | 0.8 | 0.8 | 0.8 | 0.8 |
| dropout | 0.5 | 0.5 | 0.5 | 0.5 | 0.5 | 0.5 |
| readout | SUM | SUM | SUM | SUM | SUM | SUM |

The shared hyperparameter settings of ExpC*-1, ExpC-$s$, CombC* and CombC on all 12 targets of QM9: batch sizes = 64; lr = 0.0001; step size = 30; lr decay = 0.85; readout = SUM. hidden = 256 for ExpC*-1 and ExpC-$s$; hidden = 512 for CombC* and CombC. Table 7 gives the individual hyperparameter settings of each model on each target, including the number of layers.

Table 7: Number of layers for QM9.

| | $\mu$ | $\alpha$ | $\epsilon_{homo}$ | $\epsilon_{lumo}$ | $\Delta\epsilon$ | $\langle R^2 \rangle$ | $ZPVE$ | $U_0$ | $U$ | $H$ | $G$ | $C_v$ |
|---|---|---|---|---|---|---|---|---|---|---|---|---|
| ExpC*-1,ExpC-$s$ | 5 | 4 | 5 | 4 | 4 | 4 | 4 | 5 | 4 | 4 | 4 | 4 |
| CombC*,CombC | 5 | 4 | 5 | 4 | 4 | 5 | 4 | 5 | 4 | 4 | 4 | 4 |

## I    MORE EXPERIMENTAL RESULTS

We present more results of ablation studies on OGB and QM9, which demonstrate the effectiveness of ExpandingConv, CombConv and $Re$-SUM.

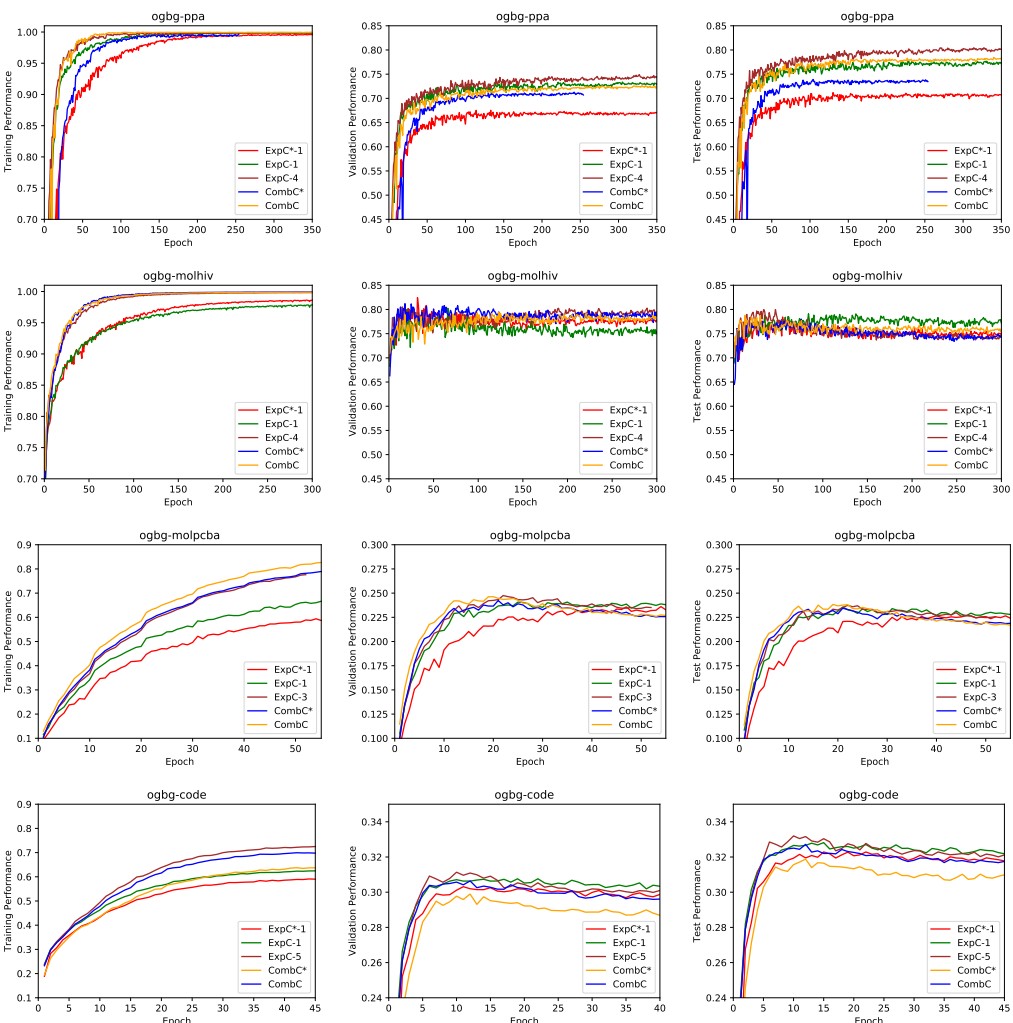

Figure 1: Learning curves on ogbg-ppa, ogbg-molhiv, ogbg-molpcba and ogbg-code.

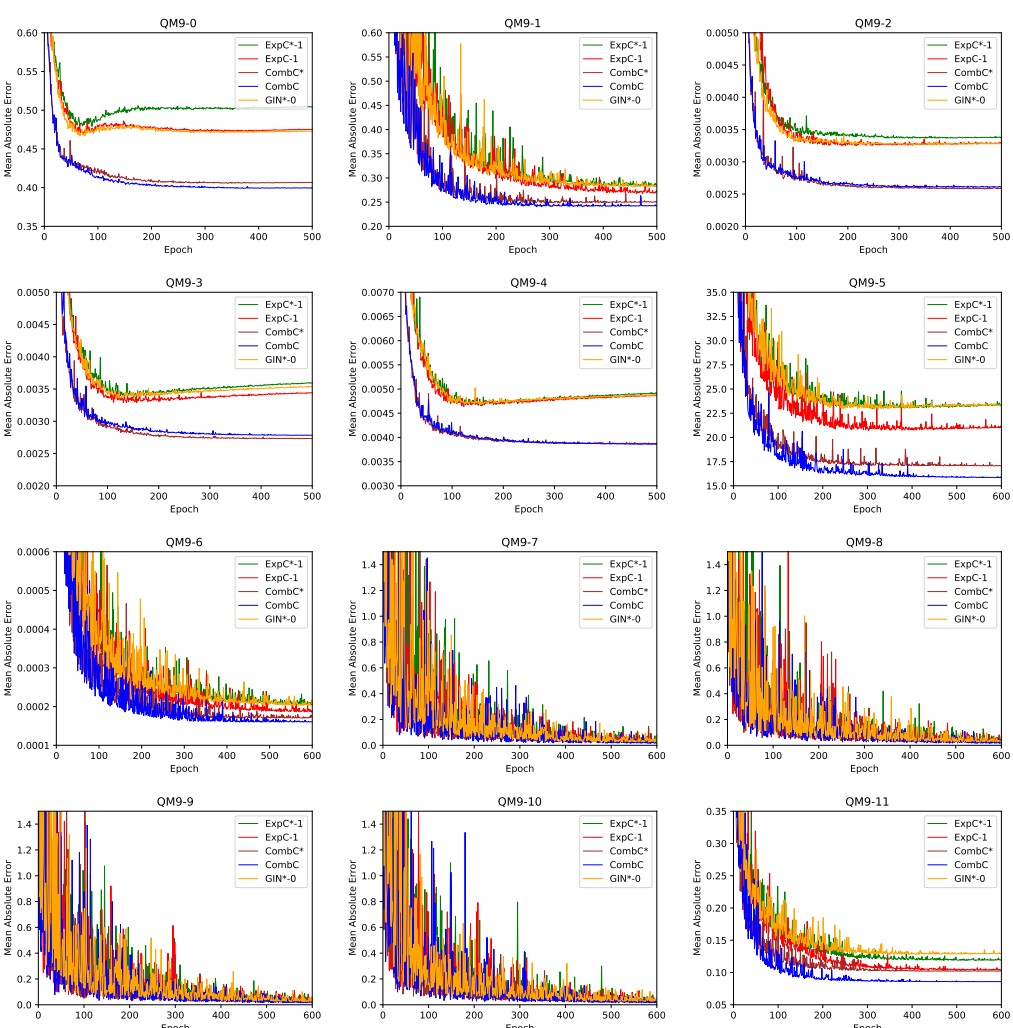

Figure 2: Effectiveness of $Re$-SUM on QM9.

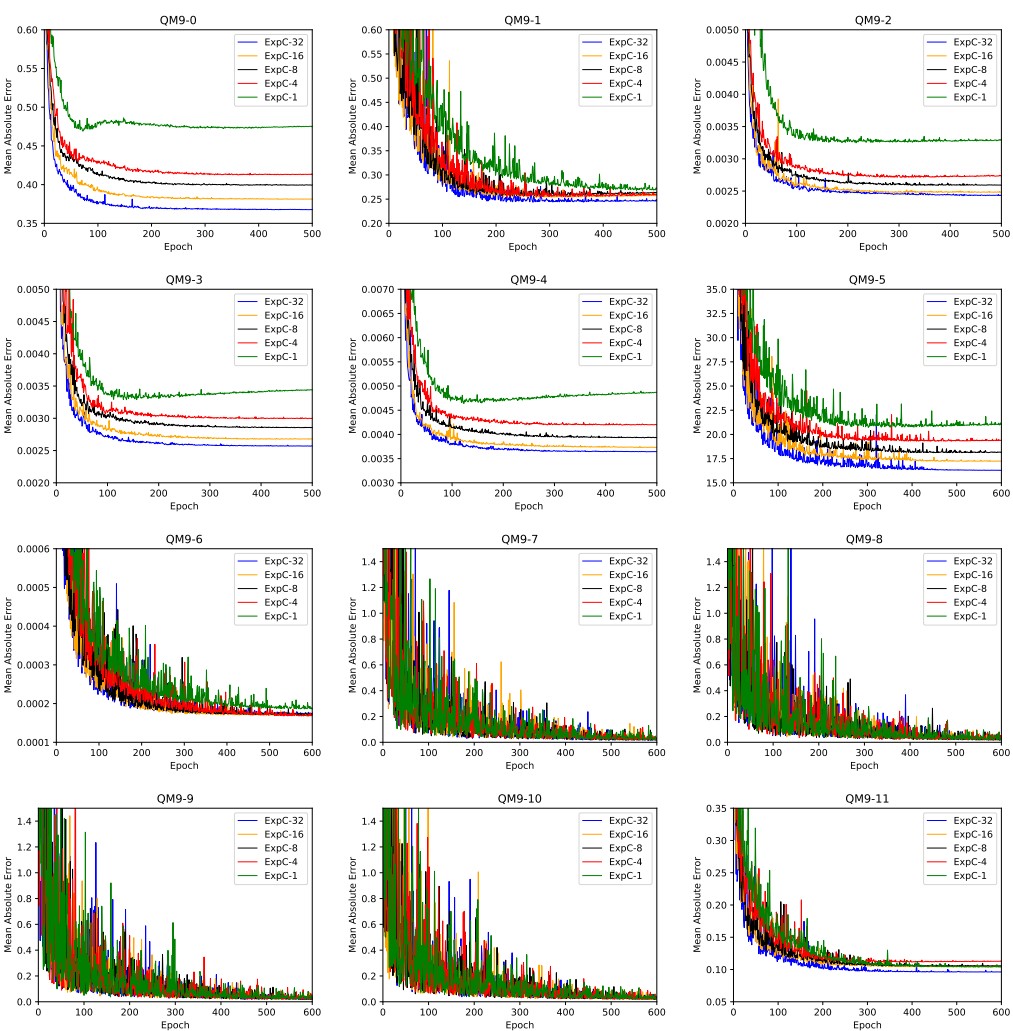

Figure 3: Effectiveness of powerful aggregators on QM9.

