# OpenReview forum: "Breaking the Expressive Bottlenecks of Graph Neural Networks"
_ICLR.cc/2021/Conference — Reject_

### Official Review · AnonReviewer4 · 2020-10-27
**The paper proposes to explore powerful aggregators to improve the expressiveness of the GNN. It is quite difficult to read the paper. Not sure that the numerical results show significant improvement.**

**Rating:** 5
**Confidence:** 3

**Review:**

Summary of the paper: The main objective of the paper is to improve the expressiveness of the GNN by exploring powerful aggregators. The requirements to build more powerful aggregators are analysed. It is closely related to finding strategy for preserving the rank of hidden features, and implies that basic aggregators correspond to a special case of low-rank transformations.

Strengths: The idea is promising. A new GNN formulation is proposed: the aggregation is represented as the multiplication of hidden feature matrix of neighbours and the aggregation coefficient matrix.

Weaknesses: The strength mentioned above (multiplication of hidden features values and the aggregation) is also a weakness: I have an impression that already known results are presented in a much more complex way. The paper is not easy to follow in general ( e.g., the sentence "The difference is that each dimension of hidden features is aggregated with an independent weighted aggregator which works like a comb".)

The paper needs to be throughly read: use \citep instead of \cite where it is necessary.

The improvements reported in the experimental section seem to be not really significant.

Questions: Could you provide an intuition for the definition of the distinguishing strength? (Section 3.1).

---

> ### Author Response · Authors · 2020-11-19
> **Our response to Reviewer 4**
>
> We sincerely thank the reviewer for the constructive comments and criticism. We will respond to each point in turn.
>
> Q1: I have an impression that already known results are presented in a much more complex way.
>
> We are not quite sure on what you mean by "already known results", and we really appreciate if you could provide relevant literatures. As far as we know, our new GNN formulation, multiplication of hidden features values and the aggregation coefficient matrix, is novel, which are recognized by other reviewers, also stated in the strengths part of your review. Our new formulation is neat, which connects the distinguishing strength with the rank of aggregation coefficient matrix. The formulation lays a solid theoretical foundation, and provides insights on how to build practical powerful aggregators. These have not been studied by existing work, although there are already many discussions on aggregator/propagation scheme/permutation invariant function [1, 2, 3].
>
> Q2: The paper needs to be thoroughly read: use \citep instead of \cite where it is necessary.
>
> Thanks for your careful reading, and we have fixed it in the new submission.
>
> Q3: The improvements reported in the experimental section seem to be not really significant.
>
> Please see the common response "We are confident in the significance of the improvements and add more dataset results".
>
> Q4: Could you provide an intuition for the definition of the distinguishing strength? (Section 3.1).
>
> Distinguishing strength of aggregation functions is defined under the concept of partial order. We explain it with the following example: To ensure simplicity, we use the familiar SUM and MEAN aggregation functions with the following 3 inputs: {1, 1, 1}, {1, 2}, {1, 1}. Then, there exist a pair of inputs which can be distinguished by SUM but not by MEAN such that SUM({1, 1, 1}) $\neq$ SUM({1, 1}), while MEAN({1, 1, 1}) = MEAN({1, 1}). Similarly, there also exist a pair of inputs which can be distinguished by MEAN but not by SUM such that MEAN({1, 1, 1}) $\neq$ MEAN({1, 2}), while SUM({1, 1, 1}) = SUM({1, 2}). Therefore, we say SUM and MEAN are incomparable under the concept of partial order (We updated the concept of incompatibility in our new submission). If we combine SUM and MEAN by concatenating their outputs, then we obtain a new aggregation function denoted by SUM $\otimes$ MEAN. Clearly, any pair of inputs distinguished by SUM or MEAN would also be distinguished by SUM $\otimes$ MEAN. Therefore, we say SUM $\otimes$ MEAN is stronger than SUM or MEAN.
>
> [1] Corso, G., Cavalleri, L., Beaini, D., Liò, P., & Veličković, P. (2020). Principal Neighbourhood Aggregation for Graph Nets.
>
> [2] Dehmamy, N., Barabási, A.-L., & Yu, R. (2019). Understanding the Representation Power of Graph Neural Networks in Learning Graph Topology. (NeurIPS).
>
> [3] Zaheer, M., Kottur, S., Ravanbhakhsh, S., Póczos, B., Salakhutdinov, R., & Smola, A. J. (2017). Deep sets. Advances in Neural Information Processing Systems, 2017-Decem(ii), 3392–3402.

---

### Official Review · AnonReviewer2 · 2020-10-28
**concerns about significance of contribution**

**Rating:** 5
**Confidence:** 3

**Review:**

**Post Rebuttal**

I thank the authors for the extensive experiments and answers. Unfortunately, I still feel that the contribution is rather marginal. I keep my score unchanged.

---

**Summary of Contributions**
The paper points at the expressive bottleneck of GNNs as the weak distinguishing strength of learned aggregators. The authors then formulate the aggregator operation via a standalone matrix, and analyze the requirements for it to have a sufficient distinguishing strength.
Then, based on the analysis, 2 aggregation schemes with enlarged strength are proposed. Also, the benefit of applying activation before aggregation is shown.

**Strengths**
- *Simplicity of analysis* - The paper presents a simple formulation to the aggregation operator via the matrix coefficients which provide an intuitive view of the requirements from the aggregator functions.
- The paper presents a general formulation for GNNs under the suggested framework, and shows how other GNNs fall into this framework.
-  *Thorough ablation study confirming theory* - the ablation study on s nicely shows the improvement in performance as s grows.
- Achieving SOTA results on some benchmarks.

**Weaknesses**
- *Contribution and comparison to Corso et. al. (2020)* - Although the paper states results regarding the requirements from strong aggregators, as very similar result has already been introduced in Corso et. al.. In that case, I would have expected to see comparison in performance as both papers tackle the same problem.
- *Comparison to GAT* - The paper shows how GAT can be formulated under the ExpandingConv formulation, raising the question, does it perform as well as the proposed methods?


**Recommendation**
The paper posses an interesting and simple view of aggregators in GNNs however I have concerns regarding the significance of contribution. Therefore, I rate it as marginally bellow acceptance threshold.

**Additional Comments**
- Inaccuracies in GCN, GAT aggregation coefficients in p.4, missing the self coefficient.

---

> ### Author Response · Authors · 2020-11-19
> **Our response to Reviewer 2**
>
> We sincerely thank the reviewer for the constructive comments and criticism. We will respond to each point in turn.
>
> Q1: Contribution and comparison to Corso et. al. (2020)
>
> This work is different from Corso et.al. (2020) in the following aspects:
>
> (i) Our work provides a systematical view on the distinguishing strength of aggregators (see the definition of distinguishing strength and Lemma 1). This makes it possible to compare the distinguishing power of any two given GNN models. To the best of our knowledge, it has not been done by other work. This formulation also provides some simple and interesting results, e.g., GIN is at most as powerful as SUM, and a 2-layer MLP is weaker than a 1-layer MLP in distinguishing strength (Consider 2-layer MLP as a composition of two 1-layer MLP and then refer Lemma 1 in our paper).
>
> (ii) From the perspective of enabling a variable number of aggregators: Enabling a variable number of aggregators is quite desired since the number of basic aggregators (such as SUM, MEAN, MAX,MIN etc) is quite limited. Corso et. al. (2020) proposed normalized moment aggregators (There are some similarities with the sum-of-power in [1]) which allow for variable numbers of aggregators. However, the proposed normalized moment aggregator is not in their PNA model, and several basic aggregators are applied instead (Multiple basic aggregator implementations are also used in [2] where they call an aggregator a propagation scheme). This raises the practical concern for normalized moment aggregators. In contrast, our proposed full rank aggregation coefficient matrix is also provably injective (see Proposition 1) but more practical. We build ExpandingConv with the aggregation coefficient matrix which is fully motivated by our analysis.
>
> (iii) Our theoretical results show more detailed requirements on aggregators in comparison to Corso et. al. (2020). For example, we analyzed the scenario for distinguishing outputs among different aggregators (Proposition 2). This ensures that different local structures are distinguishable in the case that the aggregators assigned to these local structures are different. To the best of our knowledge, this has not been studied by existing work.
>
> (v) Why no experimental comparisons with PNA proposed in Corso et. al. (2020). Our compared baseline results are taken from the original papers. Since PNA in Corso et. al. (2020) is only evaluated on their self-constructed datasets, and we did not find results of this model on public graph datasets such as OGB, QM9, and TU provided by a third party. Given the time constraints, we are not able to provide the experiment results of PNA on these public datasets, but it is definitely an interesting issue to be considered for future work.
>
> [1] Zaheer, M., Kottur, S., Ravanbhakhsh, S., Póczos, B., Salakhutdinov, R., & Smola, A. J. (2017). Deep sets. Advances in Neural Information Processing Systems, 2017-Decem(ii), 3392–3402.
>
> [2] Dehmamy, N., Barabási, A.-L., & Yu, R. (2019). Understanding the Representation Power of Graph Neural Networks in Learning Graph Topology. (NeurIPS).
>
> Q2: Comparison to GAT
>
> Please see the common response "Comparisons with GAT/multi-head GAT".
>
> Q3: p.4, missing the self coefficient.
>
> Thanks for your careful reading and we have fixed it in the new version.

---

### Official Review · AnonReviewer1 · 2020-10-28
**Strong theoretical insights into GNNs with somewhat weaker experimental results**

**Rating:** 7
**Confidence:** 3

**Review:**

The authors propose two new layers for GNNs. CombConv and ExpandingConv are motivated by the insight that a GNN is only as expressive as the rank of the matrix that represents the coefficients of the aggregation function. To arrive at this statement, the authors formalize all GNNs as being composed of three steps: 1) generation of aggregation coefficients, 2) actual aggregation of the neighbourhood, and 3) feature extraction from the aggregation. Furthermore, it is shown that current approaches have very low distinguishing strength and that CombConv and ExpandingConv, by their construction, yield higher expressive power.
The effectiveness of different components of their layers (e.g. Re-SUM, applying a ReLU non-linearity before summing when features are computed form the aggregation) are investigated in an ablation study. Additionally, proposed layers are compared with current approaches on 4 data sets.

This paper constructs an interesting theoretical analysis of GNNs and finds the bottleneck of these networks to be in the coefficient matrix of the aggregation scheme. While I was not able to check all proofs, it seems like a solid mathematical analysis. What I find somewhat sobering is the experimental section. First, I am surprised that you only compare your method on four data sets and that you miss some reported by your comparison partners (such as IMDB, REDDIT, PROTEINS, etc.). I understand the Graph Kernel data sets are smaller, however, they’ve been used in comparable papers before. Second, while the title and the theoretical analysis promises much higher distinguishing strength, I am surprised that your performance gains are good but not outstanding. I wonder if you could construct a synthetic data set in which you can show how you break the expressive bottleneck in practice. Would it make sense to compute the rank of M for different networks (including yours ((it is not guaranteed it is $s$, right?)) and plot it as a function of predictive performance?
In Eq. 4 you start with summing over all neighbours of $v$ for each dimension of $W$, could you comment on how this relates to summing over the subset of the neighbours? I am not sure, I understand what you mean with the “subset of neighbours in each dimension”.

Minor language hiccups:

•	P. 4 2nd sentence below equations: “is the function [that computes] node degrees” and same for the function “that computes” the hidden features?

•	P. 4 second to last paragraph: second sentence suiable -> suitable

•	First sentence in 3.3: Is that what you wanted to say?

•	Same page “We use […] aggregation coefficient matrices [as shown by?] Luan et al.”

•	Page 6: last sentence before “Comparisons with multi-head GAT“: “this can be explained [by the fact] that […]”.

•	Page 7: Paragraph “Effect of powerful aggregators”: Third sentence: “We config[ure]”

---

> ### Author Response · Authors · 2020-11-19
> **Our response to Reviewer 1**
>
> We sincerely thank the reviewer for the constructive comments. We will respond to each point in turn.
>
> Q1: More datasets experiments.
>
> Thanks for the suggestion. We have added experimental results on ogbg-molpcba (therefore, including all 4 graph-level predictions in OGB), 3 TU datasets (COLLAB, RDT-B, RDT-M12 ), and the results on these datasets also showed the improvements of our models.
>
> Q2: Gains are good but not outstanding. Construct a synthetic data set and evaluate on it.
>
> For the performance gains, please see the common response "We are confident in the significance of the improvements and add more dataset results". It is really a brilliant idea to construct a synthetic dataset, and then better show the performance improvements.  Our current submission is more focused on evaluating the performance on public datasets, and we would like to leave this nice idea for our future work.
>
> Q3: Would it make sense to compute the rank of M for different networks (including yours ((it is not guaranteed it is s, right?)) and plot it as a function of predictive performance?
>
> We really appreciate this constructive suggestion. Exactly as pointed out, the rank of $M$ is not necessarily $s$. In our ablation study (Table 1 in the paper), we evaluate the performance of our models by increasing $s$, and generally a better performance is achieved for a larger $s$ (probably a higher rank).  However, for the same $s$, it is difficult to configure different ranks, since the parameters in $M$ are learned. To compare with other GNN models, the performance differences are not only determined by the aggregation (or implicitly by the rank of $M$), but also depend on other factors, e.g., the feature extraction stage. Therefore, to show the predictive performance vs. rank, we need to keep other factors the same, which is not easy to control. But it is definitely a promising idea to investigate and we'd like to work on it in our future work.
>
> Q4: Transition from summing over all neighbors to summing over the subsets of the neighbors.
>
> This is achieved by applying ReLU before SUM, or the $Re$-SUM mechanism presented in the paper.  We can use a simple example to explain this process. Considering the set of neighbors {1, 2, 3, 4, 5}. In the aggregation step, each of them first multiplies with the corresponding aggregation coefficient such as {1x0.2, 2x-0.4, 3x0.7, 4x-0.1, 5x0.6}={0.2, -0.8, 2.1, -0.4, 3}. Then each element is processed by ReLU such that ReLU({0.2, -0.8, 2.1, -0.4, 3})={0.2, 0, 2.1, 0, 3}. Finally, summing them together, we have SUM({0.2, 0, 2.1, 0, 3})=SUM({0.2, 2.1, 3}). It is actually a weighted sum of the subset {1, 3, 5}. This can be used as a sampling approach, whose effectiveness is validated in our ablation study in Table 1.
>
> We have fixed language problems in the new submission. Thanks again for your informative comments.

---

### Official Review · AnonReviewer3 · 2020-10-30
**Limited context, but useful theoretical framing of distinguishing power of GNN aggregators**

**Rating:** 6
**Confidence:** 3

**Review:**

Summary:
	This paper explores the representation power of graph neural networks. Unlike recent work on choosing among simple aggregation functions or combinations thereof, the authors here recognize that these aggregators are the bottleneck in the representation power and generalize simple aggregator functions commonly used in literature to an aggregation coefficient matrix. The paper supports this construction theoretically and also proposes two aggregators that satisfy the rank-preservation requirement for more expressive (distinguishing) GNNs.

Strengths:
* The theoretical results are strong in proving the bottleneck of aggregators (Lemma 1) and clearly contextualize popular existing methods into this result.
* Formulating aggregation in terms of a product with coefficients (indexed by a permutation) allows for framing existing methods and to connect the representation power with the rank of the matrix of coefficients.

Weaknesses:
* The study of the expressiveness of GNNs is a very popular topic right now and not enough context is provided about related work on this topic and other approaches, mainly focusing on GIN and GAT in the development and while a few other GNNs are considered in the experimental results, they are not discussed or explained enough.

Recommendation:
By framing the aggregation in terms of coefficients, the paper provides interesting connections between the rank of these coefficient matrices and the distinguishing power of GNNs.  While analysis and explanation of experiments is extremely limited, the theoretical developments are interesting and novel enough to narrowly recommend publication. Meanwhile, I do think the paper should undergo a reorganization to add more details about related work in the study of expressiveness of GNNs and analyze experiments more thoroughly.


Other comments and clarification needed:
* It’s not quite clear what it means for aggregators to be incomparable (top of page 3), ie what does it mean for the relative strength to “not exist”? This point could be clarified with an additional sentence in this “Distinguishing strength” paragraph.
* The text on page 4 before Proposition 2 states that “… different M corresponds to different local structures. Therefore, the aggregation results of different aggregators must be different. However, it is not satisfied by existing GNNs.” This statement should be explained more and supported.
* The details of the ExpandingConv layer overwhelm the paper. The details fo equation 4 and related text can be moved to an Appendix and explained in the main text at a higher level. This would free up space to add context of the paper’s contribution and more properly address the experiments. In the end, the ExpandingConv formulation is a refinement on multi-head GAT.

---

> ### Author Response · Authors · 2020-11-19
> **Our response to Reviewer 3**
>
> We sincerely thank the reviewer for the constructive comments. We will respond to each point in turn.
>
> Q1: Not enough context is provided about related work.
>
> [1, 2] first discussed the expressive power of GNNs and showed that MPNN framework is at most as powerful as 1-WL test. [2,3,4,5] developed GNNs with expressive power in analogy to k-WL test. [6, 7] studied expressive power problems on node-level predictions which is related to over-smoothing problems of GNNs [8]. Our work is different from these existing work. We first theoretically showed that the aggregation functions with weak distinguishing strength lead to expressive bottlenecks on graph-level predictions. We then reformulate the aggregation in general GNN models with the aggregation coefficient matrix and then bring the connections of the distinguishing strength with the rank of aggregation coefficient matrix. We also proposed an implicit dimension-wise sampling mechanism called $Re$-SUM which is motivated by breaking the distinguishing strength of basic aggregators and also different from existing explicit neighbor sampling methods [9, 10]. We carefully updated related work in the revision.
>
> Q2: Analysis and explanation of experiments is extremely limited.
>
> We extended the experimental section with more datasets, baselines and the result analysis in the new submission. Please see the common response "Summary of revisions" and "We are confident in the significance of the improvements and add more dataset results" for more information.
>
> Q3: It’s not quite clear what it means for aggregators to be incomparable (top of page 3).
>
> We have updated our statement of incomparability (under the concept of partial order) in the new submission. Briefly, the incomparability of two aggregators $f_{aggr1}$ and $f_{aggr2}$ means that there exist multisets $x_1$, $x_2$ that can be distinguished by $f_{aggr1}$ but not by $f_{aggr2}$. Meanwhile, there also exist $x_3$, $x_4$ that can be distinguished by $f_{aggr2}$ but not by $f_{aggr1}$.
>
> Q4: The statement '… different M corresponds to different local structures ...' should be explained more and supported.
>
> We are sorry for the unclear statement and have updated in the revision. Briefly, to fully distinguish different local structures, we should make sure that their aggregated representations are different. But we know that $M$ is the mapping of local structures, and then different $M$ means that the corresponding local structures are different. Therefore, the aggregation results of different $M$ must be different. Otherwise, different local structures are indistinguishable.
>
> [1] Xu, K., Jegelka, S., Hu, W., & Leskovec, J. (2019). How powerful are graph neural networks? 7th International Conference on Learning Representations, ICLR 2019, 1–17.
>
> [2] Morris, C., Ritzert, M., Fey, M., Hamilton, W. L., Lenssen, J. E., Rattan, G., & Grohe, M. (2019). Weisfeiler and Leman Go Neural: Higher-Order Graph Neural Networks. Proceedings of the AAAI Conference on Artificial Intelligence, 33, 4602–4609. https://doi.org/10.1609/aaai.v33i01.33014602
>
> [3] Chen, Z., Villar, S., Chen, L., & Bruna, J. (2019). On the equivalence between graph isomorphism testing and function approximation with GNNs. 1–19.
>
> [4] Pan Li, Yanbang Wang, Hongwei Wang, and Jure Leskovec. Distance encoding–design provably
> more powerful gnns for structural representation learning. arXiv preprint arXiv:2009.00142,
> 2020b.
>
> [5] Clement Vignac, Andreas Loukas, and Pascal Frossard. Building powerful and equivariant graph
> neural networks with message-passing. arXiv preprint arXiv:2006.15107, 2020.
>
> [6] Oono, K., & Suzuki, T. (2019). Graph Neural Networks Exponentially Lose Expressive Power for Node Classification. 1–37.
>
> [7] Luan, S., Zhao, M., Chang, X.-W., & Precup, D. (2019). Break the Ceiling: Stronger Multi-scale Deep Graph Convolutional Networks. (NeurIPS), 1–16.
>
> [8] Li, Q., Han, Z., & Wu, X. M. (2018). Deeper insights into graph convolutional networks for semi-supervised learning. 32nd AAAI Conference on Artificial Intelligence, AAAI 2018, 3538–3545.
>
> [9] Yu Rong, Wenbing Huang, Tingyang Xu, and Junzhou Huang. Dropedge: Towards deep graph
> convolutional networks on node classification. In International Conference on Learning Representations, 2019.
>
> [10] Pushkar Mishra, Aleksandra Piktus, Gerard Goossen, and Fabrizio Silvestri. Node masking: Making
> graph neural networks generalize and scale better. arXiv preprint arXiv:2001.07524, 2020.

---

### Official Review · AnonReviewer5 · 2020-11-06
**Some interesting insights, clarity can be improved, weak experiments**

**Rating:** 6
**Confidence:** 4

**Review:**

The work presents a framework to categorize GNN aggregators based on their distinguishing strength. It connects the distinguishing strength to the rank of the aggregation coefficient matrices. Based on the findings, the authors present two GNN layers, ExpandingConv and CombConv, and evaluate them on some graph data sets.

Strengths:
- The paper is mostly well written.
- The formalization of distinguishing strength for aggregators is interesting and novel.
- The relation to the rank of coefficient matrices is a valuable insight and gives a new perspective to the effectivity of multi-head GAT.

Weaknesses/Questions:
- The paper lacks clarity in presentation / technical soundness, up to a point where I am not able to understand some details:
	- It is not clear what it means if f_aggr1 <= f_aggr2 "does not exist" (definition of incomparability)
	- In Lemma 1(ii), the formalization with "or" is confusing. It should probably be "f_aggr1 ⊗ f_aggr2 >= f_aggr1 or f_aggr1 ⊗ f_aggr2 >= f_aggr2".
	- I suggest to call the feature matrix for all nodes H, not h with specific indices, to avoid confusion. Currently, the index, or the lack of it, makes it either a matrix or a vector. Sometimes, h seems to be defined as the matrix containing features from all neighbors of u (as in the GCN definition). At other places, it seems to be just the vector of node u (Equation 3).
	- Page 5, rank(r) < min(rank(M), rank(h)) should probably be <=, otherwise rank(r) = rank(h) can't be achieved.
	- How can P be found? In order to construct it, a canonical order has to be defined and pi needs to be extracted from h. In fact, the permutation invariance does not seem to be relevant for the proposed approaches, since they both process all neighbors individually, have a sum in the end and do not impose an order by e.g. concatenation. Can the authors clarify, why this part is needed?
	- The set of results RES() is not clearly defined. The output of the aggregation is a matrix, not a set.
	- In general, Proposition 2 with the set formulations seems to be out of context and needs at least to be discussed.

- The experiments are not sufficient and do not show significant improvements:
	- Some strong competitors are left out in comparions: E.g. GAT and multi-head GAT.
	- The approach is only validated on a small set of data sets. Since the authors already use OGB, I wonder why only three data sets have been chosen. To me this looks like those three were cherry-picked.
	- Even on the small number of data sets, the results are mediocre.

- It is unclear why CombConv is proposed. It needs more discussion. In the case of CombConv, the aggregation coefficients have rank 1, similar to a lot of other existing GNN operators, isn't that right? This would mean that CombConv disgards the main argument of the work (having rank > 1) in trade for efficiency.


All in all, I recommend to reject the paper in its current state. While there seems to be some novel insight in this work, which might be of interest to the community, I think the paper needs to be improved in (1) clarity of presentation and (2) experiments. Issues with (1) can maybe be fixed within the rebuttal period. I am not sure about (2). It probably depends on the actual performance of the proposed operator.

---

> ### Author Response · Authors · 2020-11-19
> **Our response to Reviewer 5**
>
> We sincerely thank the reviewer for the informative comments and criticism. We will respond to each point in turn.
>
> Q1: It is not clear what it means if $f_{aggr1}\preceq f_{aggr2}$ "does not exist" (definition of incomparability).
>
> Incomparability here is under the concept of partial order. We have given a more clear statement of incomparability in the new submission: If there exist multisets $x_1$ and $x_2$ such that $f_{aggr1}(x_1)\neq f_{aggr1}(x_2)$, $f_{aggr2}(x_1)=f_{aggr2}(x_2)$, and there also exist $x_3$ and $x_4$ such that $f_{aggr1}(x_3)=f_{aggr1}(x_4)$, $f_{aggr2}(x_3)\neq f_{aggr2}(x_4)$, we say $f_{aggr1}$ and $f_{aggr2}$ are incomparable.
>
> Briefly, the incomparability of two aggregators $f_{aggr1}$ and $f_{aggr2}$ means that there exist multisets $x_1$, $x_2$ that can be distinguished by $f_{aggr1}$ but not by $f_{aggr2}$. Meanwhile, there also exist $x_3$, $x_4$ that can be distinguished by $f_{aggr2}$ but not by $f_{aggr1}$.
>
> Q2: In Lemma 1(ii), the formalization with "or" is confusing. It should probably be ...
>
> Thanks for your careful reading. We have fixed it in the new submission.
>
> Q3: Page 5, rank(r) < min(rank(M), rank(h)) should probably be <=, otherwise rank(r) = rank(h) can't be achieved.
>
> Thanks for your careful reading. You are right. We have fixed it in the new submission.
>
> Q4: How can P be found? In order to construct it, a canonical order has to be defined and pi needs to be extracted from h ...
>
> $P$ is only used for explaining the permutation invariance of our formulation and is not needed in the actual computations as you said. We gave this explanation because when considering the aggregation coefficients as a matrix, one may have the impression that the order of columns in this matrix is fixed and therefore raise a concern on the satisfaction of permutation invariance.
>
> Q5: The set of results RES() is not clearly defined. The output of the aggregation is a matrix, not a set.
>
> Yes, the output of the aggregation is a matrix. $\textrm{Res}(f_{M})$ denotes the set of all outputs of $f_{M}$, which is the set of matrices. We have updated the statement in the new submission.
>
> Q6: In general, Proposition 2 with the set formulations seems to be out of context and needs at least to be discussed.
>
> We have updated the statement of this part in the new submission. Briefly, we explained why the aggregation results of different $f_{M}$ should be different in our formulation. Then, we explained that Proposition 2 brings the requirements (preserving the rank of the aggregation coefficient matrix) for ensuring different results of different $f_{M}$.
>
> Q7.1: Experimental issue: Some strong competitors are left out in comparisons: E.g. GAT and multi-head GAT.
>
> We have added comparisons with GAT/multi-head GAT in our new submission. Please see the common response "Comparisons with GAT/multi-head GAT".
>
> Q7.2: Experimental issue: The approach is only validated on a small set of data sets. Since the authors already use OGB, I wonder why only three data sets have been chosen. To me this looks like those three were cherry-picked.
>
> There are only 4 graph-level prediction datasets in OGB. We have added ogbg-molpcba in our new submission, and then our models are validated on all 4 OGB graph-level datasets.. We further added the result on 3 TU datasets.
>
> Q7.3: Experimental issue: Even on the small number of data sets, the results are mediocre.
>
> Please see the common response "Summary of revisions" and "We are confident in the significance of the improvements and add more dataset results".
>
> Q8: It is unclear why CombConv is proposed. It needs more discussion ...
>
> CombConv is not motivated by preserving the rank of aggregation coefficient matrix like ExpandingConv. CombConv aggregates each dimension of hidden features within each independent subspace (also known as each independent weighted aggregator). Hopefully, this can improve distinguishing strength and better capture relevant structures in comparison to existing methods that aggregate all dimensions with the same aggregators. Anyway, this method is not fully motivated by theoretical insights compared with ExpandingConv.

---

### Author Response · Authors · 2020-11-19
**Comparisons with GAT/multi-head GAT**

First, we'd like to say that GAT/multi-head GAT is designed for node-level predictions, while our models focusing on the GNN expressiveness are more intended for graph-level predictions. Therefore, to the best of our knowledge, we are not aware of any GAT evaluations on graph-level tasks. But since reviewers are interested in the GAT/multi-head GAT performance as compared with our models, we will present the key differences between them.

### Experiment comparisons

The following table shows the experiment comparisons between multi-head GAT and our ExpandingConv on graph-level predictions (for detailed experiment settings, please see our new submission).

|                      | ogbg-ppa_________                | ogbg-molhiv_________             | ogbg-molpcba_________            | ogbg-code                        |
| -------------------- | :------------------------------- | -------------------------------- | -------------------------------- | -------------------------------- |
| multi-head GAT       | NA                               | 75.81                            | 20.10                            | 31.10                            |
| ExpandingConv (Ours) | $\textbf{79.76}\pm\textbf{0.72}$ | $\textbf{77.99}\pm\textbf{0.82}$ | $\textbf{23.42}\pm\textbf{0.29}$ | $\textbf{33.18}\pm\textbf{0.17}$ |

The results of multi-head GAT are evaluated by ourselves since there is no available GAT evaluations on graph-level predictions. We can see that our models have a better performance. We cannot obtain the results of multi-head GAT on the large and much time consuming dataset ogbg-ppa within the limited rebuttal period (hopefully, we can obtain the results before the rebuttal period ends).

### Analysis

We explain the performance differences from the perspective of the proposed three-stage representation. To achieve good performance, the methods should perform well on both aggregation and extraction stages. Specifically, in the aggregation step, they should distinguish different local structures as much as possible; in the extraction step, they should be powerful to extract the desired structural patterns on aggregated neighbor representations. Based on these insights, let us consider the performance of multi-head GAT and ExpandingConv on graph-level predictions.

**Is the aggregation in multi-head GAT and ExpandingConv equally powerful?** As we showed in the paper, the aggregation in multi-head GAT can be reformulated with an aggregation coefficient matrix, and the distinguishing power is decided by its rank. Therefore, the key is whether the rank of this matrix is big enough. However, multi-head GAT uses LeakyReLU, which is detrimental for preserving the rank (Different nonlinear units have different contributions for preserving the rank, the empirical results of LeakyReLU are provided in [1]). Meanwhile, the usage of softmax makes the rank analysis even harder. In contrast, ExpandingConv is well-motivated: ExpandingConv uses Tanh because it has been shown better in preserving the rank [1].

**Is the extraction in multi-head GAT and ExpandingConv equally powerful?** Multi-head GAT uses a 1-layer MLP, which may suffer from the representation power limitations in obtaining the desired extraction functions. ExpandingConv uses a 2-layer MLP and its representation power is guaranteed according to the universal approximation theorem.

**Our implementation of ExpandingConv includes $Re$-SUM mechanism but GAT does not.** Our extensive ablation studies showed the effectiveness of $Re$-SUM mechanism.



[1] Luan, S., Zhao, M., Chang, X.-W., & Precup, D. (2019). Break the Ceiling: Stronger Multi-scale Deep Graph Convolutional Networks. (NeurIPS), 1–16.

---

### Author Response · Authors · 2020-11-19
**We are confident in the significance of the improvements and add more dataset results**

For the significance of the improvements, we would like to advise the reviewers to refer to the OGB leaderboard [1]. The following table taken from the OGB leaderboard shows the comparisons of our models with SOTA models. The results of our methods on ogbg-molpcba are added in the new submission. There are only 4 graph-level prediction datasets in OGB, and we evaluate our models on all of them.

|                      | ogbg-molhiv_________              | ogbg-molpcba_________            | ogbg-ppa_________                | ogbg-code                        |
| -------------------- | --------------------------------- | -------------------------------- | -------------------------------- | -------------------------------- |
| GCN                  | 76.06 $\pm$ 00.97                 | 20.20 $\pm$ 00.24                | 68.39 $\pm$ 00.84                | 31.63 $\pm$ 00.18                |
| GIN                  | 75.58 $\pm$ 01.40                 | 22.66 $\pm$ 00.28                | 68.92 $\pm$ 01.00                | 31.63 $\pm$ 00.20                |
| SOTA                 | $\textbf{78.80}\pm\textbf{00.82}$ | 22.66 $\pm$ 00.28                | 77.12 $\pm$ 00.71                | 31.63 $\pm$ 00.18                |
| ExpandingConv (Ours) | 77.99 $\pm$ 0.82                  | 23.42 $\pm$ 0.29                 | $\textbf{79.76}\pm\textbf{0.72}$ | $\textbf{33.18}\pm\textbf{0.17}$ |
| CombConv (Ours)      | 77.15 $\pm$ 1.32                  | $\textbf{23.63}\pm\textbf{0.23}$ | 77.81 $\pm$ 0.76                 | 32.76 $\pm$ 0.15                 |
| VS GCN               | +1.93                             | +3.43                            | +11.37                           | +1.55                            |
| VS SOTA              | -0.81                             | +0.97                            | +2.64                            | +1.55                            |

We can see that ExpandingConv and CombConv achieve significant improvements and rank 1st on 3 out of all 4 graph-level datasets, even compared with the latest SOTA models. Especially,  ExpandingConv can be extremely powerful on densely connected graphs in ogbg-ppa.

Compared with standalone models, our models have consistently better performance. Please also note that some baselines are recently proposed and also under review. Meanwhile, the current SOTA approaches on the leaderboard are generally the combinations of several techniques, such as virtual node, data augmentation, etc. These techniques can also be incorporated into our models, leading to further performance improvements.

We further added the results on 3 TU datasets (COLLAB, REDDIT-BINARY, REDDIT-MULTI-12K) in our new submission to address the insufficient dataset concerns. Results are shown in the following table. We can see that our models achieve consistently better performance compared with SOTA models.

|                      | COLLAB____________               | RDT-B____________               | RDT-M12                          |
| -------------------- | :------------------------------- | ------------------------------- | -------------------------------- |
| DGK                  | 73.09 $\pm$ 0.25                 | 78.04 $\pm$ 0.39                | 32.22 $\pm$ 0.1                  |
| PSCN                 | 73.76 $\pm$ 0.50                 | 86.30 $\pm$ 1.58                | 41.32 $\pm$ 0.42                 |
| AWE                  | 73.93 $\pm$ 1.94                 | 87.89 $\pm$ 2.53                | 39.20 $\pm$ 2.09                 |
| GIN                  | 80.2 $\pm$ 1.9                   | 92.4 $\pm$ 2.5                  | NA                               |
| GraphSAG             | 68.25                            | NA                              | 42.24                            |
| DiffPool             | 75.48                            | NA                              | 47.08                            |
| CapsGNN              | 79.62 $\pm$ 0.91                 | NA                              | 46.62 $\pm$ 1.9                  |
| PPGN                 | 80.16 $\pm$ 1.1                  | NA                              | NA                               |
| ExpandingConv (Ours) | $\textbf{82.10}\pm\textbf{1.60}$ | 92.2 $\pm$ 1.87                 | $\textbf{49.91}\pm\textbf{1.75}$ |
| CombConv (Ours)      | 81.90 $\pm$ 1.75                 | $\textbf{92.5}\pm\textbf{1.69}$ | 49.02 $\pm$ 1.21                 |

[1] https://ogb.stanford.edu/docs/leader_graphprop/

---

> ### Comment · AnonReviewer5 · 2020-11-19
> **SOTA models?**
>
> I thank the authors for the additional effort put into the experiments.
>
> I have one further question about the results: The table with ogbg results has a SOTA row and in the text it is stated that this is the best result that has been obtained on the given task. When I take a look at the leaderboards, the numbers of the best approaches differ strongly from the ones in the table.
> In case of ogbg-molhiv, ogbg-molpcba and ogbg-code, the highest scoring models are better than the SOTA numbers presented here (and also better than the proposed method). The numbers for ogbg-ppa differ, too, but the proposed approach still outperforms it.
>
> I would ask the authors to elaborate on those differences.

---

> > ### Author Response · Authors · 2020-11-20
> > **Comparison rules**
> >
> > We are really sorry for the misunderstanding. For the comparisons with SOTA approaches, we compare the performance of standalone GNN layers without additional components (e.g., virtual node, data augmentation proposed in FLAG) to ensure a fair comparison. Meanwhile, the SOTA approaches on the leaderboard are generally the combinations of several techniques, e.g., DeeperGCN+ virtual node + FLAG.
> >
> > The comparison rules:
> >
> > ExpandingConv/CombConv                                            VS        DeeperGCN/GIN etc (reported in the above table)
> >
> > ExpandingConv/CombConv + virtual node                  VS        DeeperGCN/GIN etc + virtual node
> >
> > ExpandingConv/CombConv + virtual node + FLAG    VS        DeeperGCN/GIN etc + virtual node + FLAG

---

### Author Response · Authors · 2020-11-19
**Summary of revisions**

We thank the reviewers for their comprehensive and thoughtful comments. We carefully revised our submission based on these reviews, hoping our revisions have addressed the reviewers' concerns.

The biggest change is that we extended the experimental section to address experiment concerns raised by most of the reviewers. Below is a list of changes:

- Add the results on ogbg-molpcba dataset. Then, our models are evaluated on all 4 graph-level predictions of OGB.
- Add the results on 3 TU datasets (COLLAB, REDDIT-BINARY, REDDIT-MULTI-12K)  and compare our models with competitive baselines.
- Add the comparisons with multi-head GAT. For reviewers who are interested in the differences between our methods and multi-head GAT, we also suggest you to see the common response "Comparisons with GAT/multi-head GAT".
- Clarify the significance of improvements. For reviewers who have a concern on the improvement significance, you are welcomed to check the common response "We are confident in the significance of the improvements and add more dataset results".

---

### Decision · Program_Chairs · 2021-01-07
**Final Decision**

**Decision:**

Reject

**Comment:**

The paper explores the representation power of GNNs, in particular, studying the bottleneck and improving expressiveness with new aggregators, which are analyzed theoretically. This issue was highlighted in previous works, but the merit of this paper is a constructive analysis.

The reviewers were overall not enthusiastic and  raised a few concerns:
- Not enough context is provided about related work, in particular, the early work of Corso et. al.
- Insufficiently convincing experiments

While the authors provided an elaborate rebuttal and extended the experimental section to address experiment concerns raised by most of the reviewers, the final evaluation was still lukewarm. Given that the conference has a very high bar and there have been many very good submissions on graphs, we find the paper not quite above the bar and hence have no choice but to recommend rejection with a heavy heart. The authors should be commended on their efforts and are encouraged to seek publication elsewhere.